# Above-below surface interactions mediate effects of seagrass disturbance on meiobenthic diversity, nematode and polychaete trophic structure

Francisco J.A. Nascimento [1]*, Martin Dahl[1], Diana Deyanova[1], Liberatus D. Lyimo[2], Holly M. Bik[3], Taruna Schuelke[3], Tiago José Pereira[3], Mats Björk[1], Simon Creer [4] & Martin Gullström[1]

Ecological interactions between aquatic plants and sediment communities can shape the structure and function of natural systems. Currently, we do not fully understand how seagrass habitat degradation impacts the biodiversity of belowground sediment communities. Here, we evaluated indirect effects of disturbance of seagrass meadows on meiobenthic community composition, with a five-month in situ experiment in a tropical seagrass meadow. Disturbance was created by reducing light availability (two levels of shading), and by mimicking grazing events (two levels) to assess impacts on meiobenthic diversity using high-throughput sequencing of 18S rRNA amplicons. Both shading and simulated grazing had an effect on meiobenthic community structure, mediated by seagrass-associated biotic drivers and sediment abiotic variables. Additionally, shading substantially altered the trophic structure of the nematode community. Our findings show that degradation of seagrass meadows can alter benthic community structure in coastal areas with potential impacts to ecosystem functions mediated by meiobenthos in marine sediments.

[1] Department of Ecology, Environment and Plant Sciences, Stockholm University, Stockholm, Sweden. [2] School of Biological Sciences, University of Dodoma, Box 338, Dodoma, Tanzania. [3] Department of Nematology, University of California—Riverside, 900 University Avenue, Riverside, CA 92521, USA. [4] Molecular Ecology and Fisheries Genetics Laboratory, School of Biological Sciences, Bangor University, Bangor LL57 2UW, UK. *email: francisco.nascimento@su.se

Feedback between above- and below-surface components of soil and sediment ecosystems are a vital mechanism controlling biodiversity and ecosystem processes[1]. Anthropogenic pressure can directly affect above-surface communities, by changing community composition, resource distribution patterns, or habitat structure, which in turn can have strong effects on below-surface biota[2,3]. On the other hand, below-surface communities have an important function in organic matter mineralization and can create feedbacks that benefit above-surface communities[1,4]. Although linkages between above and below-surface habitats in driving ecosystem structure and function in terrestrial ecosystems has received considerable attention[1,2], much remains unknown about such interrelationships in marine coastal systems.

Similar to terrestrial ecosystems, plants in marine habitats provide a highly complex spatial environment with several niches for different species[5]. Seagrasses are an example of such plant communities that encompass some of the most productive habitats in marine ecosystems[6], providing a number of high-value ecosystem services[7]. Marine plant species are recognized to be autogenic ecosystem engineers shaping the shallow coastal environment through multiple and complex pathways[8]. The physical structures of seagrasses can modify local hydrodynamics and sedimentary habitats, thereby having a large controlling effect on subsurface environments by altering sediment granulometry, stabilizing sediments, storing atmospheric $CO_2$, trapping detritus, and providing a wide range of food sources that support a high diversity of consumers[9].

The abundance and diversity of below-surface metazoan consumers in marine sediments is dominated by meiobenthos (microscopic benthic invertebrates between 0.04 and 1 mm in size)[10]. Meiobenthic communities play an important role in benthic ecosystem processes[11–13]. In seagrass beds, meiobenthos are often characterized by high densities and biomass, possessing short life cycles and high turnover rates[14] that often translate into high secondary production[15]. Although the importance of seagrasses for epiphytic invertebrate biodiversity (invertebrates associated with seagrass blades and leaves) has been well documented[16], their effects on the meiobenthos in the sediment are not as well understood[17–19], in part due to the practical difficulties in large-scale studies focusing on a taxonomically hyperdiverse groups such as meiobenthos[20]. The application of high-throughput sequencing (HTS) approaches to the study of meiobenthos can considerably improve our understanding of the ecological patterns and environmental drivers of biodiversity in marine sediments[21,22], including in seagrass beds, by allowing biodiversity assessments of microscopic metazoans at a scale and with coverage previously unfeasible[20]. Nevertheless, to our knowledge no study has looked at meiobenthic diversity in seagrass beds using HTS.

Seagrass habitats and their productive below-ground communities are highly vulnerable to anthropogenic stress as they are often located in areas contiguous to intense human activities[23]. As a result, seagrass habitats have been declining worldwide due to anthropogenic activity[24]. Increased eutrophication, and sedimentation, resulting in light reduction and decreased photosynthesis, are among the principal anthropogenic disturbances to seagrass ecosystems[25]. Light reduction has multiple negative effects on seagrass plants, spanning from reduced growth and loss of biomass[26] to lower carbohydrate storage in plant rhizomes[27,28]. An additional important source of disturbance in seagrass beds comes from increased fishing pressure. The removal of predatory fishes such as wrasses (Labridae), snappers (Lutjanidae), and emperors (Lethrinidae)[29] can disturb the balance between herbivory and seagrass production and potentially induce cascading effects in these ecosystems[30]. Although, grazing is a vital process for controlling fast-growing epiphytic algae in eutrophic systems[31], release of grazers like sea urchins from predation can provoke intense grazing events that consume considerable amounts of seagrass above-surface biomass[32,33]. High densities of sea urchins and consequent overgrazing of seagrasses have been more frequently reported in the last few decades[32,33] and can have enduring impacts on above-ground seagrass biomass[32], with potential important knock-on effects for sediment properties[34] and the structure and function of benthic fauna communities. Studies on the impacts of human-induced disturbances on above-surface communities and linkages to below-surface diversity in marine systems are scarce. As meiobenthos mediate important benthic ecosystem processes, it is crucial to understand how indirect effects of eutrophication and overfishing-induced changes on plant above and below-ground biomass affect meiobenthic communities. Such an understanding is vital to predict future impacts on marine ecosystem structure and function[35].

Here, we address this important knowledge gap with a 5-month field experiment, where we manipulated seagrass plots in a *Thalassia hemprichii* meadow, and used HTS to assess impacts of shading and simulated grazing on: meiofauna species richness and evenness metrics (alpha diversity); variations in meiofauna community composition (beta-diversity) following the framework described by Anderson et al.[36]; and lastly nematode and polychaete trophic structure. The seagrass plot manipulations included two independent variables (shading and clipping) each with two levels (high and low). We used shading to mimic the effects of reduced light availability to seagrasses due to eutrophication and/or sedimentation, and simulated a high-intensity grazing event due to herbivores being released from predation. Herbivory was simulated by clipping of shoots to mimic two different levels of grazing pressure. The experimental design was used to test the following two hypotheses: shading causes a reduced seagrass root- and rhizome biomass with potential feedback effects on meiobenthic diversity, community, and trophic structure; and secondly, continued grazing causes a decrease of seagrass above-ground biomass that leads to a reduction in sediment stability and intensified erosion of the sediment surface, also with indirect effects on meiobenthic diversity, community and trophic structure. Our findings indicates that disturbance of *T. hemprichii* meadows can substantially change meiobenthic community composition and trophic structure of nematodes and polychaetes in coastal ecosystems.

## Results

**HTS data output**. The Illumina Miseq dataset of eukaryotic 18S ribosomal RNA (rRNA) amplicons generated a total of 10,320,000 raw paired-end reads from 24 samples, resulting in a total of 6,180,945 quality-filtered reads after read merging and primer trimming, which led to an average of 257,539 sequences per sample (minimum-83,262; maximum-360,378). Clustering at 96% operational taxonomic unit (OTU) similarity produced 14,106 different OTUs (minimum cluster size > 2 reads), of which 9034 OTUs were from metazoan taxonomic groups. Accumulation plots of number of OTUs vs. number of reads for each sample are presented in Supplementary Information (Supplementary Fig. 1).

**Taxonomic composition**. The percentage of OTUs belonging to metazoan groups was high for all seagrass treatments (on average 86, 80, 87, 87, and 86% in Control (CTRL), High clipping (HC), High shading (HS), Low clipping (LC), Low shading (LS), respectively), and highest in the unvegetated treatment with 96% (Supplementary Fig. 2), confirming that sieving and density

extraction is an effective way to isolate metazoan organisms as found in previous works[37]. The OTUs assigned to non-Metazoan Eukaryotes were excluded from the remaining analysis. Nematodes and copepods were the most abundant metazoan taxa in all treatments comprising ~40–70% of all relative abundance, followed by polychaetes, gastrotrichs and platyhelminths (Fig. 1a). Supplementary Data 1 presents a list of all OTUs, its taxonomic classifications and sequence counts.

At a meiobenthos group level there was an effect of treatment in the relative abundances of OTUs belonging to Nematoda (PERMANOVA, pseudo-$F_{5,18}$ = 13.9, $p$ = 0.001) and Copepoda (PERMANOVA, pseudo-$F_{5,18}$ = 4.9, $p$ = 0.004). Relative abundance of nematode OTUs were significantly higher in the Unvegetated treatment than in the CTRL (PERMANOVA, pseudo-$F_{5,18}$ = 13.9, $p$ = 0.028), while the opposite was found for copepods (PERMANOVA, pseudo-$F_{5,18}$ = 4.9, $p$ = 0.03). Within the nematodes, there were differences among treatments in relative abundances of its taxa (Fig. 1b, c). CTRL presented a significantly higher relative abundance of nematodes belonging to the order Monhysterida than in unvegetated plots (PERMANOVA, pseudo-$F_{5,18}$ = 8.6, $p$ = 0.029) and Chromadorida (PERMANOVA, pseudo-$F_{5,18}$ = 4.8, $p$ = 0.027). On the other hand, relative abundances of Desmodorida nematodes were significantly lower in the CTRL when compared to the unvegetated treatment (PERMANOVA, pseudo-$F_{5,18}$ = 31, $p$ = 0.029- Fig. 1b). At the nematode genus level there was a conspicuous difference in dominance between the seagrass plots (CTRL, HS, LS, HC, and LC) and the Unvegetated treatments. While the former were dominated by *Molgolaimus* and Monhysterids nematodes (PERMANOVA, pseudo-$F_{5,18}$ = 6.1, $p$ = 0.002, pseudo-$F_{5,18}$ = 29, $p$ = 0.001, respectively) the latter treatment was dominated by nematodes of the genus *Catanema* (PERMANOVA, pseudo-$F_{5,18}$ = 64.3, $p$ < 0.001- Fig. 1c).

**Differences among treatments in alpha-diversity**. Alpha-diversity metrics showed the same general trend for all three metrics we analyzed: observed number of unique OTUs and the ACE and Shannon index (Fig. 2). There was a significant effect of treatment on observed unique OTUs (PERMANOVA, pseudo-$F_{5,18}$ = 3.9 ; $p$ = 0.01), which was lower in Unvegetated than in any other treatments, except HC. No additional significant differences in observed unique OTUs were found between the manipulated seagrass treatments (HC, HS, LC, and LS) and the CTRL. The same pattern and effect of treatment was seen for ACE (PERMANOVA, pseudo-$F_{5,18}$ = 4.8; $p$ = 0.003) and Shannon indexes (PERMANOVA, pseudo-$F_{5,18}$ = 4.6; $p$ = 0.01). Again, both these metrics were significantly lower in the Unvegetated treatment, but not in any of the pairwise comparisons between CTRL, HC, HS, LC, and LS.

**Meiofauna beta-diversity differences among treatments**. Figure 3 shows an NMDS ordination of samples based on meiobenthic community structure across all treatments. The PERMANOVA (adonis, pseudo-$F_{5,18}$ = 2.0, $p$ = 0.001) analysis revealed a significant effect of treatment in meiobenthic community composition. A pairwise comparison performed with the *pairwise.perm.manova* function showed significant differences in meiobenthic community composition between CTRL and all other treatments (PERMANOVA, $p$ = 0.02 for CTRL vs. HS, $p$ = 0.05 for CTRL vs. LS, $p$ = 0.04 for CTRL vs. LC) except HC (PERMANOVA, $p$ = 0.09 for CTRL vs. HC). A principal coordinates analysis (PCoA) with UniFrac distance was also performed, showing a similar pattern (adonis, pseudo-$F_{5,18}$ = 4.6, $p$ = 0.001, Supplementary Fig. 3). Differences in community composition between the CTRL and all other treatments

(HC, HS, LC, LS, and Unvegetated) were mostly driven by turnover, and this pattern was constant for all comparisons (Supplementary Fig. 4). There was also a difference among treatments in community beta-diversity as measured by average distance to centroid using the altGower distance (betadisp, PERMDISP, pseudo-$F_{5,18}$ = 2.4, $p$ = 0.039) (Fig. 4). Average distance to centroid in the CTRL treatment was significantly higher from all other treatments with the exception of HC, indicating that the disturbances simulated in our experiment had a significant effect on meiobenthic community beta-diversity (betadisp, PERMDISP, $p$ < 0.02 for all pairwise comparisons between CTRL and HS, LS, and LC). A significant difference in average distance to centroid was also observed when comparing the LS and LC treatments (Fig. 4, betadisp, PERMDISP, $p$ = 0.0007).

Regarding the relationship between meiobenthic community structure and environmental variables, the BIOENV analysis showed that the environmental variables that best explained differences in meiobenthic community composition included both abiotic sediment variables (sediment C:N ratio, sediment %C content) and seagrass related biotic variables (rhizome biomass, community metabolism and N in plants) (Table 1 and Fig. 5). How each of these variables varied among treatments is presented in supplementary information (Supplementary Fig. 5). The canonical correspondence analysis (CCA) analysis showed that 43% of the total constrained inertia of the final selected model was explained, with the three retained environmental variables, sediment C:N ratio, N content in plant and C in rhizome, showing significant associations with community composition in the seagrass treatments ($R^2$ = 0.79 $p$ = 0.001, $R^2$ = 0.67 $p$ = 0.004, and $R^2$ = 0.76 $p$ = 0.001, respectively).

**Trophic composition of nematodes and polychaetes**. The abundance of trophic groups of nematodes and polychaetes was different among treatments. With regards to the nematodes, there was a significant effect of shading on the abundance of OTUs with taxonomic assignments corresponding to selective deposit feeder nematodes. The abundances of these OTUs were significantly higher in both the HS and in the LS treatment than in the CTRL (Fig. 6b, d, $p_{(DESeq2)}$ = 0.0008 and $p_{(DESeq2)}$ = 0.001, for LS vs. CTRL and HS vs. CTRL, respectively). Conversely, the abundance of OTUs of epistrate feeder nematodes were lower in the two shading treatments than in the controls, but this difference was only significant for the HS treatment (Fig. 6b, d, $p_{(DESeq2)}$ = 0.055 and $p_{(DESeq2)}$ = 0.001 LS vs. CTR and HS vs. CTRL, respectively). Significant effects of clipping on nematode trophic structure were also observed. The abundance of epistrate feeder nematode OTUs were significantly less abundant in both clipping treatments (LC and HC) than in the CTRL (Fig. 6a, c, $p_{(DESeq2)}$ = 0.021 and $p_{(DESeq2)}$ = 0.044 for LC vs. CTR and HC vs. CTRL, respectively). In addition, the abundance of non-selective deposit feeders was on average higher in the LC and HC treatments than in the CTRL, but this difference was only significant for LC (Fig. 6a, c, $p_{(DESeq2)}$ = 0.06 and $p_{(DESeq2)}$ = 0.02 for HC vs. CTRL and LC vs. CTRL, respectively). All trophic groups were significantly different between the Unvegetated treatment and the CTRL, with the predator/omnivore and non-selective nematode feeders showing an increase in abundance of OTUs in the Unvegetated treatment, whereas epistrate and selective feeding nematodes showed a decreased number of OTUs compared to the Unvegetated treatment (Fig. 6e, all $p_{(DESeq2)}$ < 0.0001).

Significant differences among treatments were also seen in the assessment of the polychaete feeding guilds. As found for nematode feeding groups, shading significantly increased the abundance of OTUs of deposit feeders polychaetes when

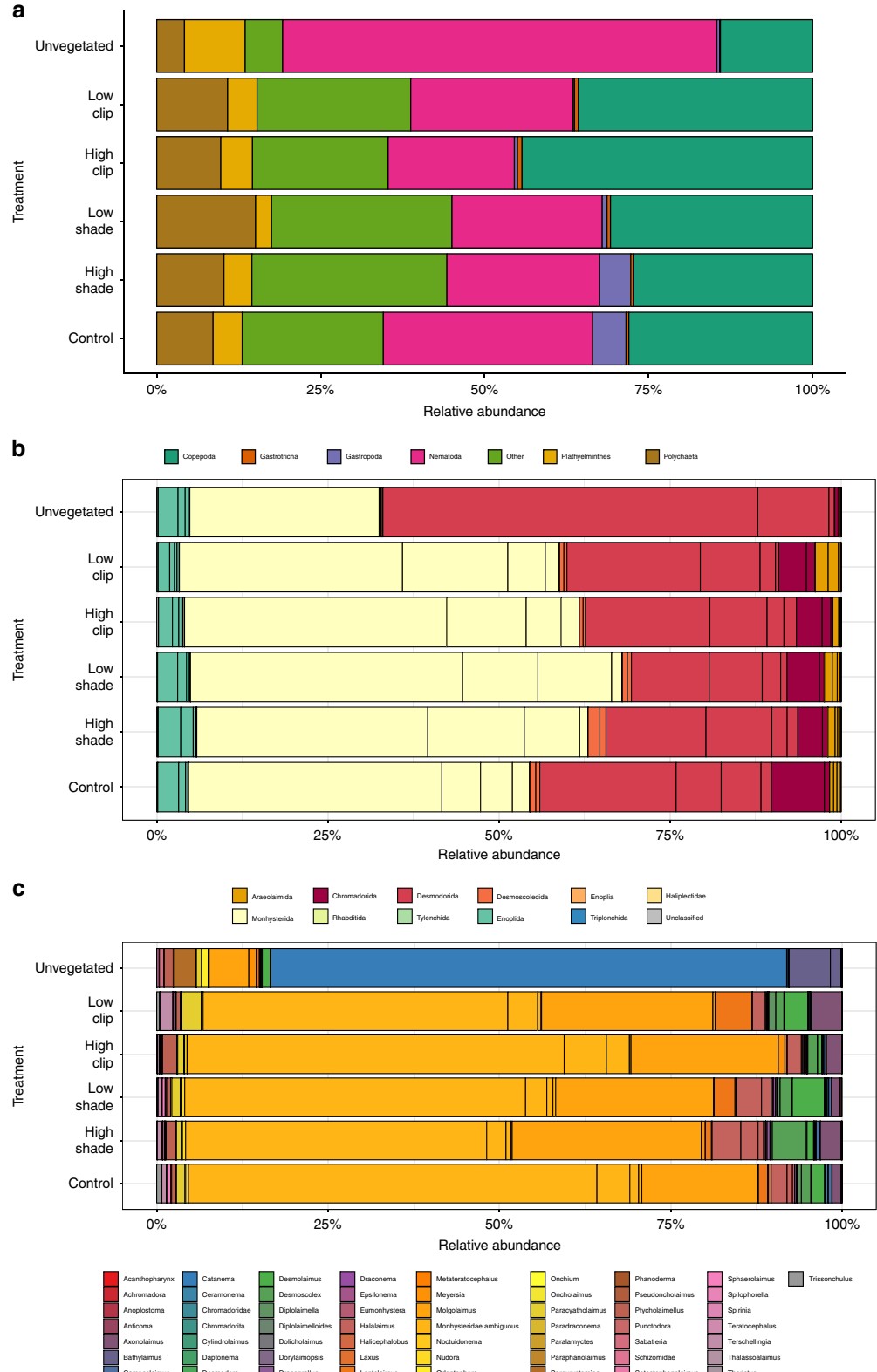

**Fig. 1** Stacked bars of the average relative abundances of 18S rRNA gene for meiobenthos in the different treatments, $n = 4$ biologically independent samples. The *y*-axis shows the treatments, and *x*-axis shows relative abundance (%) of Metazoa phyla (**a**); order in the Nematoda (**b**); and genus in the Nematoda (**c**)

compared to the CTRL, but this difference was only significant for LS, (Fig. 7a, $p_{(\text{DESeq2})} = 0.038$). In addition, significantly fewer OTUs of carnivore polychaetes were found in LS compared to the CTRL (Fig. 7a, $p_{(\text{DESeq2})} = 0.028$). No other significant differences

were found between the CTRL and the remaining manipulated seagrass treatments (HS, HC, and LC). Conversely, all polychaete feeding guilds analyzed here, with the exception of suspension feeders were significantly different in the Unvegetated treatment

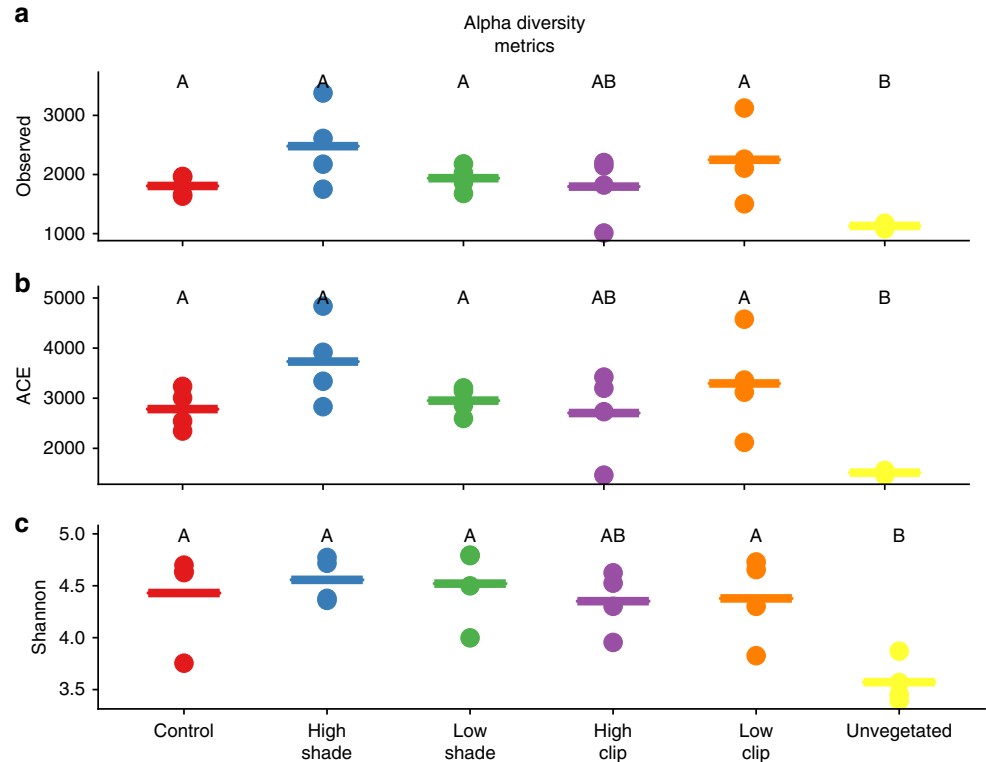

**Fig. 2** Alpha-diversity metrics for meiobenthos in the different treatments. Figure panels show: number of observed unique OTUs (**a**), ACE index (**b**), and Shannon index (**c**). Central bars represent the mean of each treatment. Different letters indicate statistically significant differences (PERMANOVA, $p <$ 0.05) based on $n = 4$ biologically independent samples

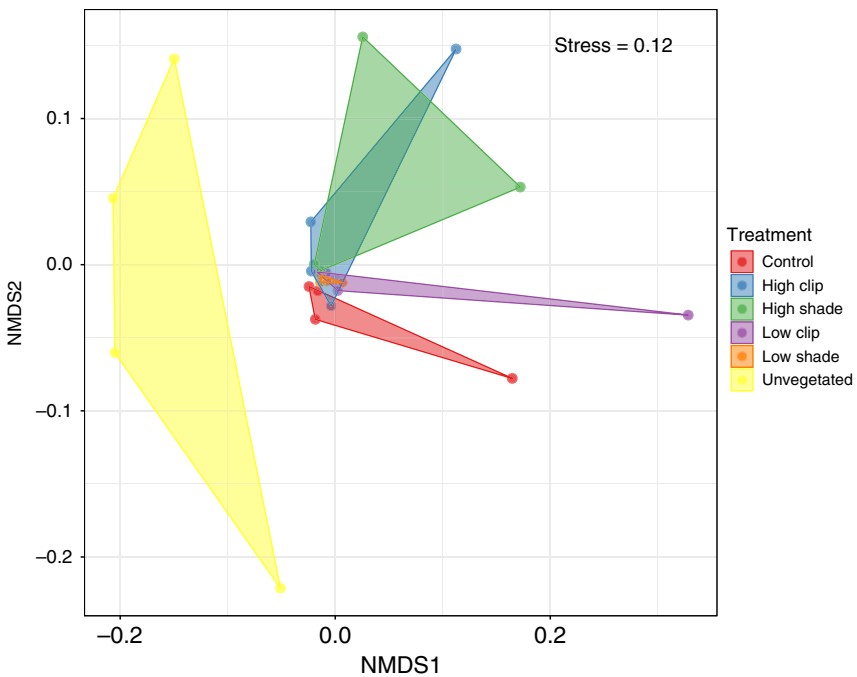

**Fig. 3** Plot of the non-metric multidimensional scaling (NMDS) analysis based on normalized OTU matrix for meiobenthos using altGower dissimilarities. Different colors represent the groupings of the different treatments

when compared to the CTRL (Fig. 7b). The Unvegetated plots had less OTUs of subsurface deposit feeders ($p_{(DESeq2)} = 0.007$) and higher abundances of OTUs in the carnivore ($p_{(DESeq2)} = 0.014$) and omnivorous $p_{(DESeq2)} = 0.043$) feeding guilds when compared to the CTRL.

## Discussion

While shading and corresponding reduced light availability did not affect meiobenthic community alpha-diversity in our study, it had a significant effect on meiobenthic community structure. Reduced light availability to seagrasses is often coupled to

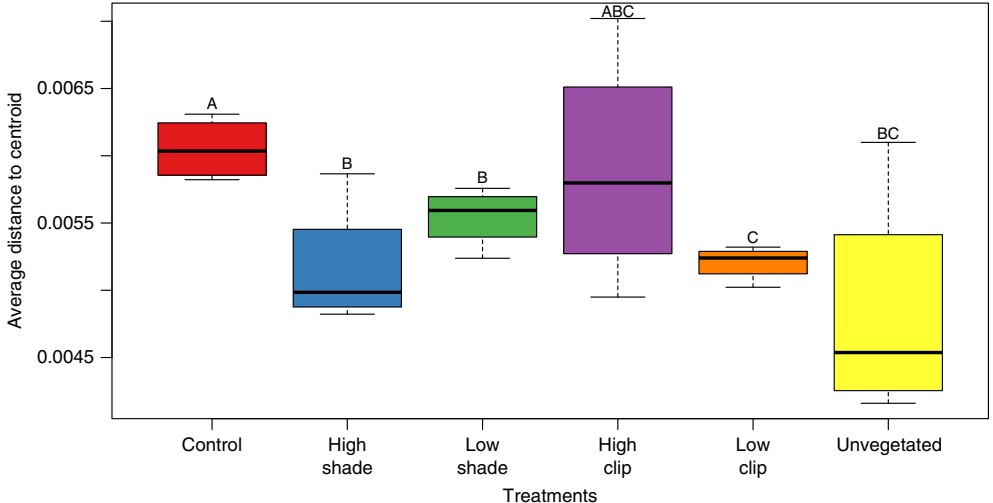

**Fig. 4** Meiobenthic community β-diversity index showing the average distance from group centroid to each observation, $n = 4$ biologically independent samples. Different letter codes indicate statistically significant differences (PERMDISP, $p < 0.05$)

**Table 1 Biota-environment (BIOENV) analysis showing the five best combinations of variables linked with the highest correlation to the meiobenthos community composition**

| No of variables | Correlation | Environmental variables |
|---|---|---|
| 7 | 0.6 | Sed C:N ratio; Bulk C in core; Sed C inorg; Rhizome biomass; NCP; N in Plant; C in rhizomes |
| 7 | 0.598 | Sed C:N ratio; Bulk C in core; Sed C inorg; Leaf biomass; Rhizome biomass; NCP; C in rhizomes |
| 6 | 0.597 | Sed C:N ratio; Bulk C in core;Sed C inorg; Leaf biomass; Rhizome biomass; NCP |
| 8 | 0.594 | Sed C:N ratio; Bulk C in core; Sed C inorg; Leaf biomass; Rhizome biomass; NCP; N in Plant; C in rhizomes |
| 6 | 0.593 | Sed C:N ratio; Bulk C in core; Sed C inorg; Rhizome biomass; NCP; N in Plant |

Correlation values represent Spearman's rank correlation coefficient. Environmental variables abbreviations: Sediment C:N ratio (Sed C:N ratio); Bulk carbon density (Bulk C in core); Sediment content in inorganic C (Sed C inorg); Rhizome biomass (Rhizome biomass); Community metabolism (NCP); Plant Nitrogen content (N in Plant); Rhizomes carbon content (C in rhizomes): Leaf biomass (Leaf biomass)

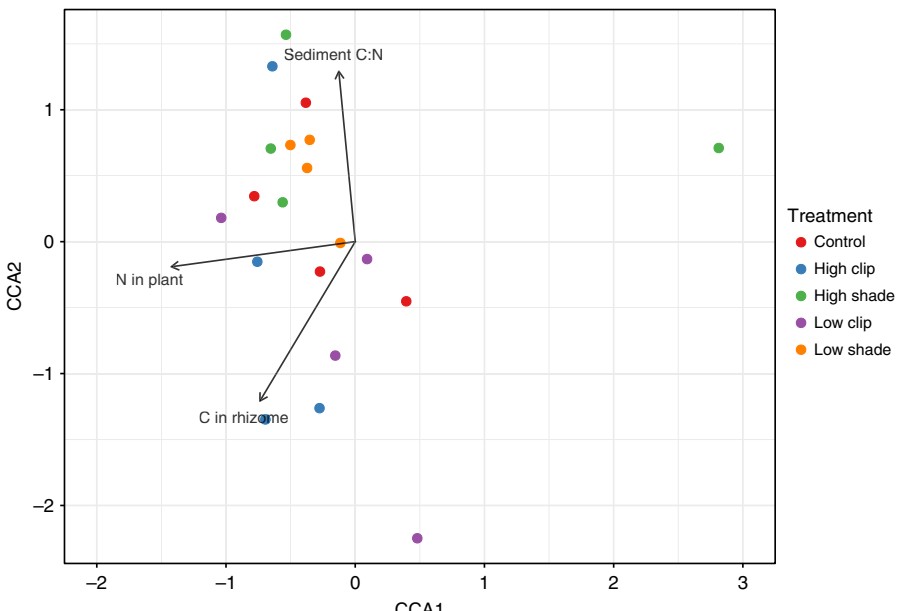

**Fig. 5** Canonical correspondence analysis (CCA) biplot showing the co-variant relationship between significant non-correlated environmental factors (see Methods) and meiobenthic community structure. Arrows are vectors representing the correlation between environmental variables and the axes

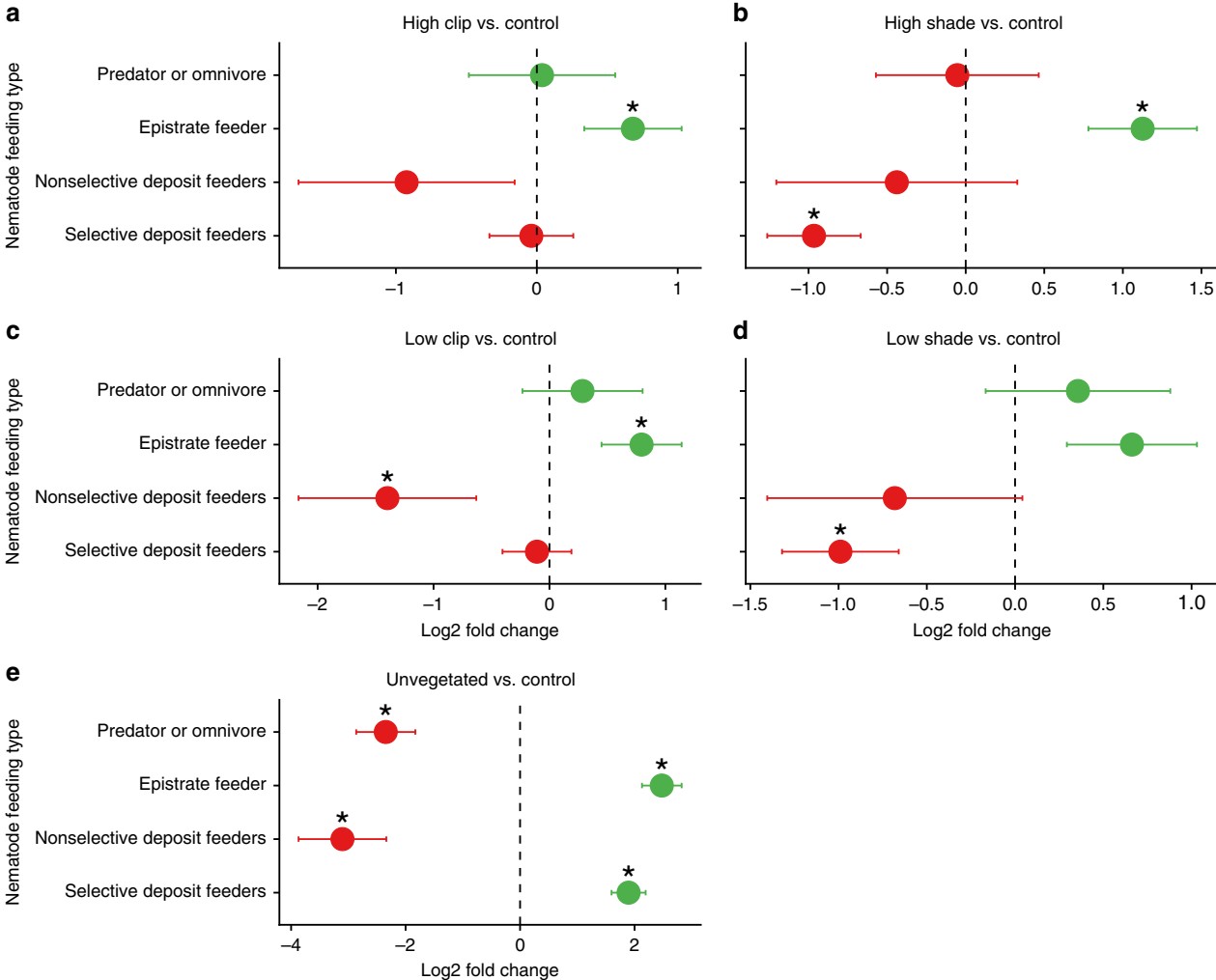

**Fig. 6** OTU abundance of nematode feeding types in the different treatments in relation to the Controls: High Clip (**a**), High Shade (**b**), Low Clip (**c**), and Low Shade (**d**), Unvegetated (**e**). The x-axis shows the $\log_2$ fold changes of the four different nematode feeding types (y-axis) calculated by the DESeq2 adjusted base mean (see Methods). A $\log_2$ fold change of > 0 (green) indicate that abundance was higher in the Controls than in the respective manipulated treatment, while a $\log_2$ fold change of < 0 (red) indicates that abundance was lower in the Controls than in the respective manipulated treatment. Asterisks show cases when differences were statistically significant ($p_{(DESeq2)} < 0.05$) and error bars represent SE, $n = 4$ biologically independent samples

eutrophication and/or increased sedimentation in seagrass beds[25]. Decreased light availability as a result of increased phytoplankton and epiphytic algae production is one of the principal mechanisms through which eutrophication impacts seagrass meadows[24,28]. Seagrasses can acclimate to reduced light regimes by decreasing above and below-ground biomass and photosynthetic activity[28,34,38], which in turn potentially shape sediment abiotic conditions for meiobenthic communities[17]. In particular, *T. hemprichii* has a comparatively well-developed root and rhizome network[39] that can confer stability to the sediment and increase its microscale complexity that favors microbial growth and diversity[40]. As such, a decrease in below-ground biomass of *T. hemprichii* could potentially impact such microscale habitat complexity and sediment characteristics for the meiobenthos.

Lower biomass and photosynthetic activity as a result of reduced light availability will cause a lower transport of oxygen from the shoots to the roots, decreasing "radial oxygen loss (ROL)" from the root-tips and thereby reduce the oxygenation of the sediment[41]. Reduction in photosynthetic rates can also lead to higher $H_2S$ levels in the sediments of disturbed seagrass meadows[41,42]. Both lower oxygen conditions and increased $H_2S$

concentrations in sediments have the potential to change meiofauna diversity and community composition[43,44]. In addition, photosynthetically derived dissolved organic carbon (DOC) has been shown to greatly stimulate the activity of microorganisms around *T. hemprichii* roots when it is transported to below-ground tissue and excreted from the root system[45]. Both disturbances here tested, shading and clipping, probably reduced the amount of DOC extruded from the roots to the sediment. As bacteria and some nematodes can utilize DOC as an energy source, these direct and indirect changes in resource availability are likely to have effects of meiobenthic community structure. Similar in situ studies have shown that shading resulted in a significant decrease in root biomass and photosynthetic activity in the HS treatments[28], and the BIOENV analysis in our study identified rhizome biomass as one of the variables that correlated with meiobenthic beta-diversity. These results suggest that a reduced microhabitat complexity could be related to the changes in meiobenthic community beta-diversity in the shading treatments.

In addition to an effect on meiobenthic community beta-diversity, we found that the relative abundances of OTUs

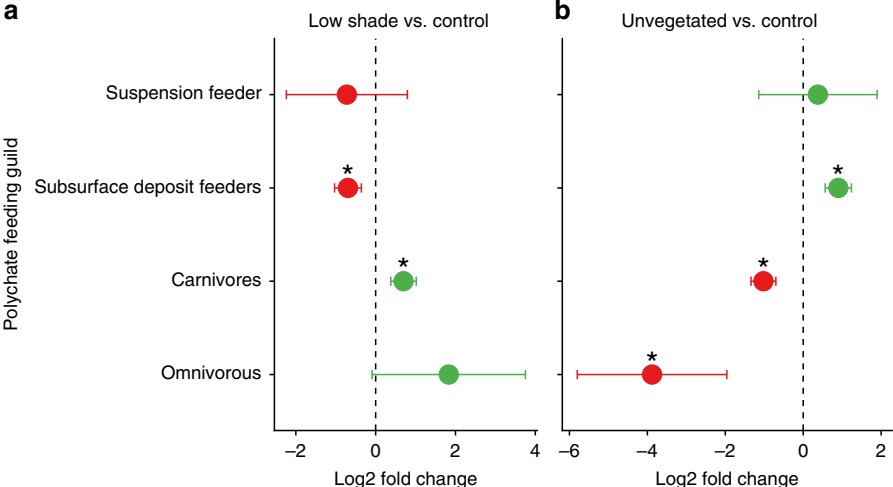

**Fig. 7** OTU abundance of polychaete feeding guilds in the different treatments in relation to the Controls: Low Shade (**a**) and Unvegetated (**b**). The x-axis shows the $\log_2$ fold changes of the four different nematode feeding types (y-axis) calculated by the DESeq2 adjusted base mean (see Methods). A $\log_2$ fold change of > 0 (green) indicates that abundance was higher in the Controls than in the respective manipulated treatment, while a $\log_2$ fold change of < 0 (red) indicates that abundance was lower in the Controls than in the respective manipulated treatment. Asterisks denote cases when differences were statistically significant ($p_{(DESeq2)} < 0.05$) and error bars represent SE, $n = 4$ biologically independent samples. No other differences were detected between CTRL and the remaining manipulated seagrass treatments

assigned to nematodes of different feeding types differed significantly between the control and the shading treatments, the latter showing a lower proportion of epistrate feeders that seemed to be replaced by selective deposit feeders. Nematodes are generally one of the most abundant metazoans in seagrass systems and associated trophic structures are determined by abiotic factors, such as grain size, sediment porosity, temperature, salinity, and food availability[10]. In our study, both temperature and salinity varied in the same way among treatments and differences in sediment porosity, and compactness did not explain changes in the trophic structure of nematodes (BIOENV, Table 1). As such, our results suggest that the reduction in OTUs of epistrate feeder and increase in OTUs of selective deposit feeders are related to changes in the food resources of these two feeding types of nematodes. Changes in food quantity and quality have been coupled to nematode trophic structure in seagrass *Posidonia oceanica* meadows[15,46] and in other coastal ecosystems[47]. We propose that shading reduced important phytoplankton food sources to epistrate feeder nematodes, as well as sedimentation in these plots, thereby decreasing the relative abundance of epistrate feeders. We expected such effects would be noticeable in the sediment Chl*a* content and net community production (NCP). A study that used the same experimental system as ours found community metabolism to be significantly lower in the HS treatment than in the CTRL plots[34]. While Chl*a* sediment content was on average higher in the controls than in the shading treatments, this difference was not statistically significant[34]. On the other hand, nematodes classified as selective deposit feeders are generally considered to depend on different food sources than epistrate feeders, as they preferentially feed on bacteria, small particulate food or dissolved organic matter. As such, selective deposit feeders would therefore not be affected by the changes microphytobenthic production and phytoplankton sedimentation. The reduced competition with other nematodes could explain the increase in selective deposit feeders. Changes in nematode trophic structure should be interpreted cautiously as recent work suggests that most nematodes in their natural environment might exhibit a certain level of generalist and opportunistic feeding behavior[48]. Nevertheless, the classification

of Wieser (1953) still provides valuable information about the feeding guilds of nematode community.

The increase in deposit feeders in the shading treatments observed in the nematode community was also seen in polychaetes (Fig. 7). Unlike what was seen with nematode feeding types, the abundance of predator polychaetes was reduced in one our shading treatments. This is in accordance with previous studies that have found an increase in dominance of polychaete deposit feeders and a decrease proportion of carnivores as an observed response to anthropogenic disturbance in benthic ecosystems[49,50]. Taken together our results clearly show an indirect effect of shading on meiobenthic community composition and trophic structure that is mediated by seagrass response to eutrophication/and or increased sedimentation. Our results suggest that the impacts of eutrophication on seagrass meiofauna community and nematode and polychaete trophic structure can at least in part be due to indirect effects mediated by the response of seagrasses to reduced light availability, and that above-belowground interactions can play an important role in mediating sediment community structure in marine ecosystems.

Clipping also produced seagrass mediated effects on meiobenthic beta-diversity, but these were less clear than what could be observed in the shading treatments. The largest impact of these manipulations on the seagrass was the continuous removal aboveground photosynthetic shoot from the replicate plots, an effect that simulates the impact of intense grazing events[51]. This loss of biomass is known to disrupt the carbon sequestration and the trapping of allochthonous organic matter, an important component of organic carbon in seagrass beds[52]. Therefore, it was expected that a loss of above-ground biomass would result in a lower accumulation of allochthonous organic matter in the clipping treatments. Indeed, Dahl et al.[34] found a lower organic carbon content in the first 2.5 cm layer of sediment of the clipping treatments in the same experimental system here reported. Organic carbon content has been shown to be one of the most important factors structuring meiobenthic communities[46] and it is likely that seagrass mediated effects on sediment carbon dynamics affected the meiofauna community structure in the clipping treatments. Indeed, BIOENV analysis found both

sediment carbon content and sediment C:N ratio correlated with changes in meiobenthic community structure. An additional notable consequence of continuous shoot biomass removal is an increased sediment erosion due to reduced capacity of shoots to decrease wave action[34]. A decreased root and rhizome biomass (significant only in the HC treatment) would also reduce sediment stability and allow for a higher degree of erosion[53], which is particularly relevant in our experimental area characterized by large tides and strong wave action[54]. This increase in tidal disturbance and sediment erosion as a result of seagrass biomass removal has been seen as a response to large grazing effects by sea urchins[55]. As such, both reduction of allochthonous organic matter trapping and increased erosion are expected to impact sediment abiotic conditions important for the structuring of meiobenthic communities. Additionally, loss of canopy can also reduce protection from predation. Macrophytes provide shelter from predation for both macro-[5] and meiobenthos[17]. It is possible that increased predation pressure contributed to the differences in meiobenthic community structure. However, we did not measure predation pressure in our experiment and are unable to confirm the connection with the data available. We expected the effects on meiobenthic community should have been more pronounced in the HC than the LC treatment. However, we found that community beta-diversity to be significantly different from the CTRL in the LC but not in the HC treatment. It is possible that the high erosion and tidal action in HC increased the variability within replicates, thereby decreasing our power to detect statistical differences. An additional explanation is that, although simulated grazing treatments can reduce the biomass of rhizomes, the root and rhizome network is still present and minimizes potential negative effects of above-ground disturbances on meiobenthic communities. It would therefore be interesting to test the effects of high clipping with higher amounts of replication.

We also anticipated changes in sediment condition in the HC treatment to affect the trophic structure of nematodes; in particular the abundance of OTUs of epistrate feeders as Dahl et al.[34] found significantly higher Chla content in HC sediments. This higher Chla content found in that study would suggest a higher microphytobenthos production as a result of a greater light availability due to the removal of seagrass above-ground biomass. However, we did not detect a higher OTU abundance of epistrate feeder nematodes in the HC treatment when compared to the CTRL but rather the opposite. It is likely that the sediment erosion and high hydrodynamics of our experimental system, would increase with lower seagrass canopy and induce the observed patterns in nematode trophic structure. Although an effect of clipping was detected on meiobenthic beta-diversity, community composition and nematode trophic structure, our results indicate that disturbance related to clipping has less pronounced effects when compared to shading.

There were clear differences between unvegetated areas and CTRL in most response metrics here studied, including meiobenthic alpha diversity, meiobenthic community beta-diversity, nematode, and polychaete trophic structure. The CTRLs had higher alpha-diversity, abundance of epigrowth feeder nematodes, and carnivore polychaetes than the Unvegetated plots. Positive effects of seagrasses on macrofauna diversity and abundance of macrofauna are well known[56,57] but regarding the less studied meiobenthos, the available literature shows contrasting results[17] and references therein. For example, Arrivillaga and Baltz[58] found no significant differences in meiobenthic abundance, species richness or diversity between sediments in tropical *T. testudinum* meadows and unvegetated sediments. Furthermore, a number of studies have shown meiobenthos abundance to be negatively correlated to seagrass cover as a result of

increased predation pressure by macrofauna on vegetated sediments[59,60]. Nevertheless, the positive effects of *T. hemprichii* for meiobenthic alpha and beta-diversity, and trophic structure were clear in our study. Seagrass cover increases the stabilization of sediments, habitat complexity, and sediment organic matter content, all of which could have positive effects on meiobenthos[17,18,61]. Our results suggest that this habitat modulation by seagrasses influenced nematode community composition. Unvegetated sediments were dominated by Desmodorida, particularly of the genus *Catanema* that seem to find unstable fluid sediments in unvegetated areas advantageous[14,18]. However, other studies have found *Catanema* to be common in seagrass areas at sediment depths deeper than the ones sampled in our experiment[18,19]. *Catanema* was replaced by *Molgolaimus* in our seagrass plots, a common nematode genus in sediments of *T. hemprichii* meadows, particularly in its top layer[18]. These seagrass plots were clearly dominated by Monhysterida, which are likely positively impacted by increased amounts of fine particles and detritus normally found in sediments in seagrass meadows[62]. Effects of seagrass on nematodes and other meiobenthos may, nevertheless, be dependent on seagrass species' composition and density and on other abiotic factors not examined here.

In summary, our results indicate that disturbance of seagrass meadows have propagating effects on meiobenthic communities that are mediated by above-below-ground interactions. Shading altered meiobenthic community composition and nematode and polychaete trophic structure to a larger dominance of deposit feeders. Such responses to shading by the meiobenthos seem to be related to reduced seagrass root and rhizome biomass reported in previous studies[28,34]. The continued simulated grazing in the clipping treatments also resulted in significant changes in meiobenthic community and trophic structure, although these were not as clear as the shading treatments. Our study suggests that such changes are connected to a decrease in above-ground biomass and intensified erosion of the sediment surface reported in previous work[34]. Since human-induced disturbances are increasing the rate of seagrass bed habitat degradation[63], it is crucial to improve our understanding of what such losses mean for the structure and functioning of benthic ecosystems. Our results highlight the complex role of above-below ground interactions in marine systems. Seagrasses function as ecosystem engineers for benthic faunal communities, and how they respond to disturbances can have significant indirect effects of meiobenthic community diversity and trophic structure. Considering that meiobenthos have important roles in benthic foodwebs[10,35] and mediate vital benthic ecosystem function[11,13], prolonged disturbances of seagrass habitats as presently seen in many coastal waters, are likely to have important cascading effects for benthic ecosystem structure and function.

## Methods

**Study area and experimental setup**. We performed an in situ experiment for 5 months (November to March 2015) in a seagrass meadow in Chwaka Bay on Zanzibar Island (Unguja), Tanzania. Chwaka Bay is a large (~50 km²) semi-enclosed bay on the east coast of Zanzibar Island with a maximum (spring tide) tidal fluctuation of 3.2 m[54]. The bay is composed by seagrass meadows (with as many as 11 seagrass species) and unvegetated bare sediment habitats[64]. Within the bay, an experimental site (06°09'S 39°26'E) was selected in the middle of a one kilometer-wide seagrass meadow dominated by *Thalassia hemprichii*; a common species in the region, as well as in tropical areas elsewhere[65]. The experimental site was located in the intertidal zone with a water depth of ~10 cm during low spring tide. Salinity was 34 in the experimental area and was measured with a multimeter Multi 340i, CellOx 325 (WTW).

The experimental design comprised six treatments; low- and high-clipping intensity treatments (LC and HC, respectively), low- and high-shading treatments (LS and HS, respectively), as well as controls of non-manipulated seagrass plots (CTRL). Unvegetated bare sediments plots were selected in an area adjacent to the manipulated plots. Four replicate plots for each treatment were placed within a 40 × 40 m experimental site using a random block design, with each plot covering

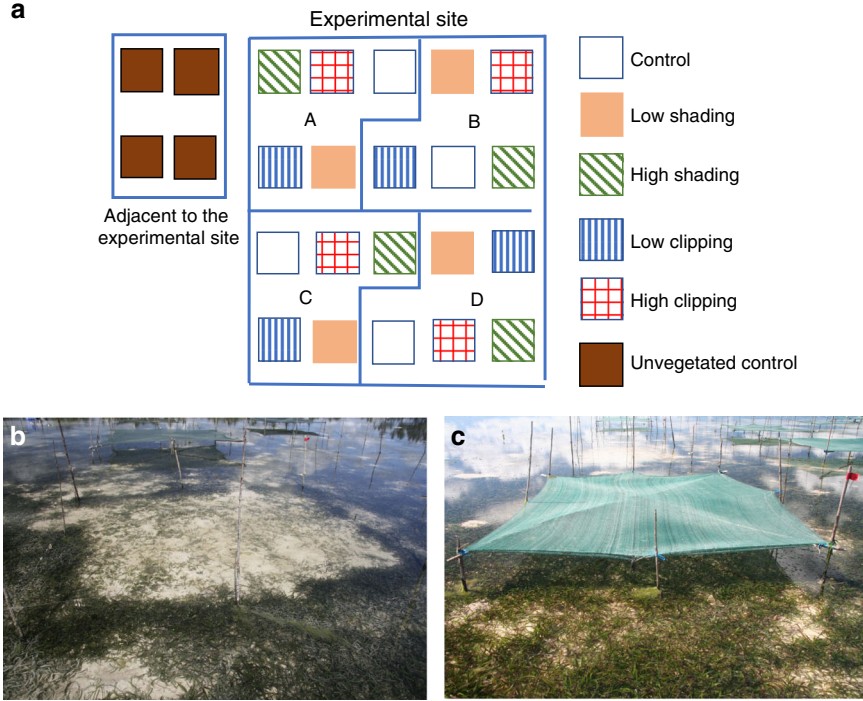

**Fig. 8 Experimental approach. a** Experimental approach displaying the randomized complete block design in our study. Different patterns correspond to the different experimental treatments (four biologically independent replicates per treatment). Letters represent replicate blocks. **b** High-Shading treatment, **c** High-Clipping treatment. Photos by Martin Gullström

10 m² [28,34] (Fig. 8). The LS and HS plots were covered with plastic semi-transparent shading nets, mounted ~40 cm above the sediment surface; the LS treatment was covered with one shading net and the HS treatment with double-shading nets. This procedure reduced the light irradiance from 470 μmol quanta m$^{-2}$ s$^{-1}$ in the seagrass control plots, to 356 μmol quanta m$^{-2}$ s$^{-1}$ in the LS treatment (a mean light reduction over day of 64% in relation to CTRL) and 307 μmol quanta m$^{-2}$ s$^{-1}$ in the HS treatment (a mean light reduction over day of 75% in relation to CTRL). A photosynthetic active radiation (PAR) Logger (Odyssey, New Zealand) was used to measure light intensity levels in LS, HS, and control plots. Each day the shading nets were cleaned of debris and fouling organisms, and the nets were replaced two times during the experiment due to natural wear. For LC and HC treatments, 50% and 100% of the original shoot biomass was removed, respectively. In the LC treatment, the shoot height was reduced by approximately half the natural shoot length (~10 cm) and in the HC treatment, the shoots were cut just above the meristematic region. The clipping was performed at a 3 to 5 day interval until 3 weeks before terminating the experiment after which no additional clipping was done.

**Sediment sampling, sample preparation, and sequencing.** After 5 months, at the termination of the experiment, the sediment of each of the 24 replicate plots was sampled with six handheld Perspex sediment cores taken from the exact same location within each of the plots. The handheld sampling units were 45 mm diameter with a surface area of 17 cm², a size suitable for sampling of microbial benthic metazoans such as meiofauna[66,67]. The top 3 cm of each core were sliced and sieved through 500 μm and 40 μm stacked sieves, pooled and preserved in 20% solution containing dimethyl sulphoxide, disodium EDTA, and saturated NaCl (DESS) before storage at 4 °C. After 2 weeks, the sediment and animals were again placed in a 40 μm sieve and rinsed thoroughly in filtered artificial saltwater (salinity 34) close to in situ salinity to remove the DESS. The meiofauna individuals were isolated and separated from the sediment particles using density extraction by washing the content of the 40 μm sieve into a 500-mL E-flask with LevasilH 200A 40% colloidal silica solution (H.C. Starck SilicaSol GmbH) with a density of 1.3 and shaken vigorously as described previously in Nascimento et al.[11]. After aeration, the solution was left to settle for 5 min. The top 100 mL of the LevasilH solution was sieved through a sterilized 40 μm sieve and rinsed thoroughly in seawater. The 40 μm sieves were then washed with 70% ethanol and autoclaved between each replicate. The density extraction procedure was repeated twice (5-min and then 30-min settling time). The extracted meiofaunal animals were then washed carefully from the sieve into a 50 mL falcon tube with a volume of Milli-Q ultrapure water that did not exceed 10 ml and frozen at −20 °C until DNA extraction.

**DNA extraction.** DNA from the meiofauna community was extracted with the PowerMax® Soil DNA Isolation Kit (MOBIO, Cat#12988), in conformity with the protocol instructions. After DNA extraction, samples were frozen at −20 °C in 3 mL of C6 solution (10 mM Tris). After this, 100 μL of each DNA extract was purified with PowerClean® Pro DNA Clean-Up Kit (MOBIO, Cat# 12997-50) and stored in 100 μL of C5 (10 mM Tris) solution at −20 °C. Before PCR amplification, all DNA extracts were standardized to a concentration of 10 ng/μL. The conservative metabarcoding primers TAReuk454FWD1 (5′-CCAGCA(G/C)C(C/T) GCGGTAATTCC-3′) and TAReukREV3 (5′-ACTTTCGTTCTTGAT(C/T)(A/G) A-3′) and Pfu DNA polymerase (Promega, Southampton, UK) were used to amplify the 18S nSSU gene region with PCR, creating fragments between 365 and 410 bp, excluding adaptors or barcodes. Each sample from the 24 replicate plots were amplified in triplicates, which were then pooled, dual-barcoded with Nextera XT index primers following a modified version of Bista et al.[68] and visualized by gel electrophoresis. The barcoded amplicons were then purified with the Agencourt AMPure XP PCR Purification kit (Beckman Coulter), quantified with Qubit (Invitrogen, USA), and pooled in equimolar quantities. The purified amplicons were sequenced in both directions on an Illumina MiSeq platform at the National Genomics Institute (NGI -Stockholm, Sweden) as a single pool comprises the 24 different samples with 24 unique index primer combinations (i.e., an index primer combination for each of the four replicates plots of our six experimental treatments).

**Bioinformatics.** Amplicon reads were demultiplexed by the sequencing facility, followed by initial data processing and quality-filtering in the QIIME 1.9.1 pipeline[69]. Paired-end Illumina reads were overlapped and merged using the join_paired_ends. py script in QIIME, followed by quality-filtering of raw reads using the multiple_split_libraries_fastq.py script with a minimum Phred quality score of 19. Unmerged (orphan) Illumina read pairs were discarded, and excluded from all downstream data analysis steps. PCR primer sequences were subsequently trimmed from merged reads using Trimmomatic version 0.32[70] (parameters used were ILLUMINACLIP:2:30:10, with all other parameters as default). Trimmed, merged reads that passed all quality-filtering steps were next subjected to open-reference OTU picking using a 96% pairwise identity cutoff, using the pick_open_reference_otus.py script in QIIME 1.9.1 (using the uclust algorithm with 10% sub-sampling, no pre-filtering, and reverse strand match enabled). All resulting singleton OTUs were excluded from the resulting OTU table outputs. Taxonomy was assigned to representative OTU sequences with the RDP Classifier[71] in QIIME (assign_taxonomy.py with a confidence threshold of 0.7), using the SILVA 119 release as a reference database[72]. OTU representative sequences were aligned with PYNAST[73] using the align_seqs.py script.

**Statistics and reproducibility**. The resulting OTU table and correspondent metadata set was imported into R v 3.4.3 and analyzed using the phyloseq[74] and vegan[75] packages. The effect of both shading and clipping on alpha-diversity metrics (observed OTUs, ACE index, and Shannon index) and relative abundance of meiofauna taxonomic groups were tested with one-way PERMANOVA with the PAST 3.24[76]. Statistical significance was defined at $\alpha = 0.05$ to cover all analyses.

Community composition was examined by first selecting and filtering metazoan OTUs and sub-sampling the OTUs counts to the lowest sample size (66,754 counts) with the *rarefy_even_depth* function in pyloseq. After Hellinger transformation, the dissimilarity between faunal assemblages in the different treatments was analyzed by non-metric multidimensional scaling (NMDS), using the altGower distance[77], and by principal coordinates analysis (PCoA) with UniFrac distance. To statistically test for the effects of treatment on community composition, we conducted a permutational multivariate analysis of variance (PERMANOVA) with the *adonis* function of the *vegan* package. The function *pairwise.perm.manova* of the RVAideMemoire package[78] was used to perform pairwise comparisons between CTRL and the remaining treatments in terms of differences in community composition. To examine differences in beta-diversity among treatments we used the community beta-diversity index[36] that is based on community OTU dissimilarity metrics and measured as average distance of each observation to the group centroid, using the *betadisper* function in the *vegan* package[75]. Pairwise differences between treatments in average distance to the group centroid were checked with the *permutest.betadisper* of the betadisp object that permutes model residuals and generates a permutation distribution of F with the null hypothesis that there is no difference in dispersion between groups. Furthermore, metrics to partition beta-diversity were utilized to calculate the relative importance of turnover and nestedness in the different treatments[79]. Beta-diversity can be divided into dissimilarity as a result of turnover, i.e., species replacement between sites or samples, and dissimilarity as a result of nestedness, species loss from sample to sample. We used the R package betapart[79] for this analysis. Additionally, a BIOENV ("biota-environment") analysis[80] was performed to explore relationships between environmental variables and meiobenthic community composition using Spearman's rank correlations. Concisely, BIOENV identifies the combination of environmental variables, that best correlated with the changes in community structure. For the analysis, we included 21 variables measured and reported in Dahl et al.[34] and Deyanova et al.[28], studies based on the same experimental system. Specifically, we used two classes of environmental variables for the BIOENV analysis. Firstly we used seagrass traits namely: net community production (NCP); leaf biomass, C, N content, and C:N ratio; rhizome biomass C, N content, and C:N ratio; root biomass, C, N content, and C:N ratio. Secondly we also used sediment variables, specifically: density, porosity, sediment %C, sediment %N, sediment C:N ratio, sediment inorganic C, and content in total hydrolysable amino acids (THAA) and Chla. The methodology used to derive these variables is described in detail in Dhal et al.[34] and Deyanova et al.[28]. Furthermore, and in order to complement the BIOENV analysis and visualize the relationships between the environmental variables and community composition, a CCA was performed with the best combination of variables identified by BIOENV as a starting point. After exclusion of the variables that had a correlation coefficient higher than 0.7 from the analysis, we used the *envfit* function of the *vegan* package to test which environmental variables were significantly correlated with meiobenthic community composition.

To investigate potential changes in nematode trophic structure, we subset our dataset to include only nematode OTUs that could be taxonomically classified to genus, in a procedure similarly applied to terrestrial nematodes[81,82]. These 644 OTUs were categorized into functional feeding groups as previously defined by Wieser[83], using nematode buccal cavity morphology to define four trophic groups: selective deposit feeders (1A), non-selective deposit feeders (1B), epistrate feeders (2A), and omnivorous-carnivorous (2B). A full list of nematode feeding type classifications for the genera used in this is available in Supplementary Data 2.

Furthermore, we investigated treatment related changes in the trophic structure of polychaetes, by subsetting the polychaete OTUs taxonomically assigned to Family (total 870 OTUs) and classifying them to relevant trophic guilds (e.g., deposit feeders, omnivore, herbivore, or predators) following Jumars[84].

To assess differential OTU abundance between the CTRL and the other treatments in nematode and polychaete trophic structure, we used the DESeq2 statistical package[85]. DESeq2 accounts for the variance heterogeneity often observed in sequence data by using a negative binomial distribution as an error distribution to compare abundance of each OTU between groups of samples[85]. All statistical tests were performed on R v 3.4.3. All statistical analysis outputs can be found in Supplementary Data 3.

**Data accessibility**. The raw sequence data have been uploaded and are available on the NCBI database with the following BioProject number: PRJNA540961

**Reporting summary**. Further information on research design is available in the Nature Research Reporting Summary linked to this article.

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

## Acknowledgements

We would like to the MEFGL staff for help with the laboratory work. F.N.'s participation in this project was supported by the Swedish Research Council (grant number 623-2010-

6616), the Swedish Research Council Formas (Future Research Leaders grant number 2016-1322), and the Lars Hierta Minne Foundation. Sequencing was performed at the National Genomics Institute, Sweden, with the support of the SciLifeLab National Project in Genomics and the Knut and Alice Wallenberg Foundation. Research activities were funded by the Swedish International Development Cooperation Agency (Sida) through the Bilateral Marine Science Program between Sweden and Tanzania and through a 3-year research project grant (SWE-2010-184). Open access funding provided by Stockholm University.

## Author contributions

F.J.A.N., M.D., D.D., L.D.L. M.B, S.C., and M.G designed the study. M.D., D.D., and L.D. L. conducted the experiment and sampled in the field. F.J.A.N. conducted the laboratory work and analyzed the data; T.S. and T.J.P. provided with bioinformatics and, with help from H.M.B. F.J.A.N. wrote the manuscript with contributions from M.D., H.M.B., M.B., S.C., and M.G., and comments by D.D., L.D.L.

## Competing interests

The authors declare no competing interests.
