## [Peer Review File · Communications Biology]

Reviewers' comments:

Reviewer #1 (Remarks to the Author):

General

-Studies on meiobenthos are traditionally time-consuming and training requirements. Although the high-throughput sequencing (HTS) had been widely used in various fauna and microorganisms, it was rarely used in meiobenthos. The field experiment in Chwaka Bay had already been studied and published on the other themes (e.g. Dahl et al. 2016, Deyanova et al. 2017), and this study is focused on meiobenthos and the changes of their community structure after the seagrass treatments.

-Personally, I am not an expert of molecular biology so I am not familiar with the high-throughput sequencing (HTS) or related analyses. I would focus on the contents of meiobenthos and seagrass and their relationships.

-The authors have good taxonomic data of nematodes from HTS, whereas there is no any detailed discussion about the alteration of nematode assemblages. For example, regarding Desmodorida, does *Catanema* in unvegetated site replace by *Molgolaimus* in other seagrass treatments? It seems like that it is not appropriate to overextend the meaning of OTUs, but the relative abundance of OTUs might still reflect some interesting variances in communities.

-It does need some minor alterations. Please find my suggestion as the following.

Introduction

60 In De Troch et al. (2001), the "benthic" and "epiphytic" meiofauna actually indicate the meiobenthos associated with unvegetated and seagrass sediments, respectively. And meiofauna are usually considered identical to meiobenthos. Therefore, it better uses "epiphytic invertebrate biodiversity" here and need to cite the other references.

78 It might be better to add the family names of the fishes, e.g. emperors (Lethrinidae), for potential misunderstanding. This sentence may need to rewrite for easy reading.

103 ...a decrease of seagrass above-ground biomass.

Results

128 If all the Mollusca in Supplementary Fig. 3-A belong to gastropods, it should denote Gastropoda.

Discussion

265 Most nematodes might exhibit a certain level of generalist in feeding. However, their body sizes and sizes of buccal cavities restrict their food resources. The classification of Wieser (1953) still provide valuable information about the feeding guilds of nematode community.

280 It seems like that the feeding guilds of nematode and polychaete have slightly different responses to seagrass treatments. Of course, the feeding groupings may not truly represent their dietary habits, but some comparison between nematode and polychaete might be interested.

311 It is also possible that your clipping treatment not affect the below-ground community as severe as above-ground one. Cutting leaves and shoots would reduce the biomass of rhizomes, but the root and rhizome network still occurs.

342 About the effects of seagrass species' composition and density on meiobenthos. You may cite De Troch et al. (2001) and Liao et al. (2016).

Liao, J.-X., Yeh, H.-M. & Mok, H.-K. Do the abundance, diversity, and community structure of sediment meiofauna differ among seagrass species? *J. Mar. Biol. Assoc. U. K.* 96, 725–735 (2016).

Methods

368 Do the unvegetated replicate plots locate outside the experimental site? Since they are not indicated in Fig. 7, please declare in Methods.

388 A surface area of 17 cm² is an appropriate size for sampling of meiofauna. Need references.

Fig. 5 and 6

The white asterisks showing significantly different are clear in the manuscript. However, it might be difficult to distinguish in a reduced-size copy. A traditional indication, e.g. asterisks beside the dots, may be appropriate here.

Reviewer #2 (Remarks to the Author):

Reviewer: Dr. Maickel Armenteros.

General comments

The manuscript addresses the ecological interactions between meiofauna and seagrass meadow features (perturbations) using an experimental approach. The research is pertinent and interesting to the generality of the marine ecology community working on meiofauna-diversity-ecosystem functioning. It covers a gap in the knowledge and provides novel and experimentally tested conclusions. Authors used a simple and well-designed experiment to test their hypotheses. They also applied modern molecular techniques to describe the meiofaunal communities. The manuscript is in general well written and focused on the goals. The statistical analysis is adequate for this sort of experimental work

I note that bioinformatics and HTS are far to be my expertise, so I did not focus on them. I present a set of criticisms and detailed comments below, but in general, positive features largely overcome the negative points, and I congratulate to the authors for so nice work.

Detailed comments

1. L1-3. Trophic structure refers only to nematodes. I suggest "... meiobenthic diversity and nematode trophic structure".

2. L19-20. Abstract. There is some knowledge about impacts of seagrass degradation on E-F; I suggest modulating this statement since it is unfair.

3. L27. Abstract. Change parameter x variable.

Introduction

4. L56. Space between number and unit.

5. L77-83. These two sentences are key, but as formulated hard to understand. I suggest rewriting, mentioning explicitly the grazers that increase activity when predator decrease. That is, more background information about the trophic links in the system.

6. L95. Change variable x treatment.

7. L99. Please, rephrase "herbivore release from predation".

8. L100-105. It is supposed that all the background information was given previously. Is suggest remove references from hypotheses to easier reading.

Results

9. L115-116. I suggest rephrase as: Accumulation curves of number of OTUs versus number of samples for the six treatments.

10. Supplementary fig. S1. Correct the caption. It is not a rarefaction curve. Rarefaction is a technique to compare the samples in reference to the smallest sample size. This is an

accumulation curve of OTU vs. number of samples.

11. L117. I suggest that supplementary fig. S3 be included as a figure in the main manuscript and not as supplementary. It is relevant information about the results of the experiment and the ability of the HTS to describe the meiofauna community.

12. L122-123. The word 'remaining' is repeated in same sentence, improve.

13. Supplementary fig. S3. Caption lacks important information. Indicate what are in the panels a, b, etc.

14. L125. Denote the table S1 as it. There are two excel files with cryptic names.

15. L128: 7 x seven.

16. L135. Alpha diversity has several definitions. I suggest to devote a couple of lines in the introduction to define what are the diversity metrics you are going to use, for inventory (\sim alpha), and variation (\sim beta) diversity components.

17. L135. Repeated words, rephrase the sentence.

18. L137. I suggest using PERMANOVA instead of ANOVA for consistency. Former based on Euclidian distance is equivalent to ANOVA, it can be used in a univariate context as well, and is more robust to asymmetry and variance heterogeneity than ANOVA. Thus, you do not have to test for ANOVA assumptions (e.g. Bartlett test) and make transformations.

19. L146. Strictly speaking: ... the ordination of samples based on community structure...

20. L147. PERMANOVA does not use F, instead pseudo-F.

21. L165. Ration x ratio.

Discussion

22. L345. I suggest adding the word summary or synthesis here. The relevance/transcendence of the study is here, and it is fine. But, I feel that some brief summary of the findings would improve this ending paragraph.

M & M

23. L357. Year?

24. L369. The phrase four replicate plots should be in a separate sentence.

25. L370. It means that each plot is 3 m x 3 m? Are the plots square or rectangular (as in fig. 7)? Please, clarify.

26. L371-383. I suggest to add in the fig. 7 some photos of the application of the treatment. E.g. How the deployed shading nets look? How look the seagrass when removed the blades? Maybe fig. 7 can be passed to supplementary.

27. L444-446. If you chose to use PERMANOVA, you do not have to use Bartlett test.

28. L447-448. Mention Tukey post hoc test in other parts of the text instead of ANOVA (see comment above).

29. L456-459. Consider invert the order of these two sentences foe easier reading.

30. L487. I suggest to change mouth x buccal cavity.

Table

31. L716-718. Table 1. Names of variables (e.g. sed., core) should be clarified in the heading (foot?) of the table.

Captions

32. L721-723. Caption fig. 1. Use metrics instead of measurement for consistency. C capital. Remove codes. Post hoc test is not an ANOVA, should be Tukey.

33. L725. Caption fig. 2. The term "beta-diversity matrix" is confusing. Ordination of the samples is based on OTU matrix using certain similarity index.

34. L729. Caption fig. 3. Should be PERMDISP instead PERMANOVA.

35. L733-737. Actually, you represented here a biplot (samples + environmental "external" variables). I would remove CCA1 and CCA2. The phrase "... co-variant relationship between significant non-correlated environmental factors" is confusing; I suggest rewrite. Arrows are vectors representing the correlation between environmental variables and the axes.

36. Supplementary Fig S4. Sorensen.

Figures

37. L764. Fig. 1. I suggest adding a measure of central position (e.g. mean or median) as a line (or bar), since statistical tests say differences in relation to mean (or centroid). It is matter of opinion, but I see too much blank space in the graphs (i.e. low information density). Larger size in

letter and symbols would improve the plot.

Reviewer #3 (Remarks to the Author):

This manuscript (MS) provides experimental evidence to support the proposition that disturbance above the sediment surface in sea grass meadows generates indirect effects on meiobenthic diversity and trophic structure below/in the sediment. The authors focused on two types of disturbance, (i) reduced light availability (high/low) and (ii) grazing (high/low) with a 5-month (or week?) experiment, assessing the effect of the aforementioned treatments on alpha and beta diversity measures as well as community trophic structure of nematodes and polychaetes.

The main conclusions were that both treatments had an effect on meiobenthic communities, primarily at the beta-diversity level, nematode trophic structure (high shading/low clipping treatments) and polychaete community structure (low shade).

This MS represents a well-executed, balanced experiment with an original (to my, albeit, limited knowledge on the subject) perspective on linking disturbance above the sediment surface to effects on infaunal taxa. The MS is generally well written, but will benefit from general grammatical improvements and increased clarity and conciseness.

The application of high-throughput sequencing (HTS) adds to the value of the MS, however the adoption of OTUs rather than Amplicon Sequence Variants (DADA2) strikes me as disadvantageous given how the former are known to overestimate richness and are characterized by higher error rates in taxonomic assignments. The Materials & Methods section for Bioinformatics is lacking important parameter details. In addition, I would think that a beta-diversity measure that includes phylogenetic information (e.g. UniFrac) would be more appropriate given that sequence data are being analysed, compared to the chosen Gower distance.

Many environmental variables were measured during these experiments (21 total), however we have no visual representation of their patterns in the different treatments.

A figure of the ones that were most highly correlated with meiobenthos community composition at minimum (BIOENV analysis) would be appropriate as currently the reader has no way of knowing the general trends.

Line-by-line comments:

Line 24: "assessed" should be "assess"

Line 56: Meiobenthos definition by Giere actually extends to 1000µm. These days meiofauna is usually considered to lie in the 32-1000µm range.

Line 96: "We used shading to simulate the effects of reduced light availability to seagrasses at levels compatible with ecologically realistic effects of eutrophication and/or sedimentation [...]". How was this ecological compatibility and relevance determined? Include reference if available with relevant irradiance levels, if not I would delete this sentence.

Line 129: "Zooming in" change to "Within"

Line 130: "There were differences". Were these statistically tested?

Line 148: "A pairwise comparison performed with PERMDISP". I am unsure if this is the most appropriate way to test the effect of treatments. This would be achieved through PERMANOVA, which requires a non-significant PERMDISP as an assumption (non-parametric equivalent of a homoscedasticity test). As far as I understand, PERMDISP will indicate which groups have unequal variance, but not which groups means differ, which is what you are trying to test. Non-parametric

pairwise comparisons in R can be completed with `pairwise.perm.manova` or the `pairwise.adonis2` function. As this is used throughout the text, it's crucial to clarify what exactly is being tested.

Line 150: "Average distance to centroid". Is this the Gower distance? If so, mention it by name.

Line 165: Correct "ration" to "ratio".

Line 172: "In regards to the nematodes" change to "With regards to the nematodes"

Line 224: "Both lower oxygen conditions ..." Include reference for this statement

Line 226: "DOC". First time this acronym is encountered so should be written in full

Line 232: "In accordance, shading resulted ...". The way this sentence is phrased gives the impression to the reader that is referring to the current study, yet a reference is included, rephrase in a manner that makes it obvious you are referring to a different study, e.g. similar shading experiments have shown ...

Line 243: "Foods sources available to the nematodes" change to "food availability" to avoid repeating "nematodes" twice in one sentence.

Line 248: "Functional classes" or feeding types? Both have been used in the MS, chose one for consistency.

Line 250: "We propose that..." This sentence needs restructuring for clarity. Microphytobenthos and diatoms are both phytoplankton so, if I understand correctly, essentially the message is that shading reduced phytoplankton production and therefore epistrate-feeding nematodes. E.g.: We propose that shading reduced important phytoplankton food sources to epistrate feeder nematodes as well as sedimentation in these plots, thereby decreasing the relative abundance of epistrate feeders.

Line 255: "NCP was in fact significantly ..." Here again the phrasing gives the impression that the conclusion is made from the current study data, but a reference is provided. Rephrase to clarify which is true.

Line 258: "The different food sources" Which ones? Or just different? Rephrase sentence for clarity

Line 266-268: "Nevertheless, because of its easy application" You just mentioned that the feeding types should be interpreted with caution and that's sufficient in my view, I would delete this entire sentence justifying using Wiesers scheme.

Line 276: "Nematode community" But also polychaetes right?

Line 289: "Indeed, this was reflected ..." Here again the phrasing gives the impression that the conclusion is made from the current study data, but a reference is provided. Rephrase to clarify which is true.

Line 301: "As a result from seagrass biomass" change to "as a result of seagrass biomass"

Line 320: "HC treatment compared to the CTRL". Here again the phrasing gives the impression that the conclusion is made from the current study data, but a reference is provided. Rephrase to clarify which is true.

Line 328-344: I think it is worth mentioning that sea grass beds can have a negative effect on meiofauna abundance, and that these effects may be species-specific. However, the MS is aimed at demonstrating the effects of the experimental treatments, for which the CTRL site is relevant. Highlighting differences between CTRL and unvegetated seems to me irrelevant here, and also

rather obvious that these would differ.

Line 370: Fig.7 illustrates 5 of the 6 treatments, where are the unvegetated blocks located?

Line 373: "Mean light reduction over day of 64% or 307 $\mu\text{mol quanta m}^{-2} \text{s}^{-1}$ " Would be more comprehensible for the reader to state what the CTRL irradiance is first (470), and then to what irradiance the LS and HS will be. LS: $470-307=163$ final irradiance or reduction of 65% ($307/470=65.32\%$) | HS: $470-356=114$ final irradiance or reduction of 75%

Line 386: "After 5 weeks" Should be 5 months correct?

Line 387: "Sediment cores taken from the same points inside the plots" What does this mean? That you took 6 replicates from the exact same location in each of the 24 plots?

Line 393: Delete 40 μm , mentioned in previous sentence. "In situ salinity" Which was what? Measured how?

Line 403: "Volume of water" What kind of water? Care was taken to keep the sieves sterile, but frequently water can be the source of contamination in samples. Did you use Milli-Q water as the final solution for the extracted animals?

Line 415: "Each sample from the 24 replicate plots" If I understand correctly, each of the 24 pools includes the triplicates of each of the 6 replicate cores per plot? Difficult for the reader to understand what happened to those 6 replicate cores, include this info for clarity.

Line 429: "Quality filtering". What was the chosen phred-score threshold? This is important information.

Line 432: "Trimmomatic" Again, what were the chosen parameters? This is important information.

Line 437: "RDP Classifier" What was the chosen confidence threshold? This is important information.

Line 445: Was data normality tested? This should also be checked.

Line 446: Delete "to examine for all ANOVA analysis", mentioned at beginning of sentence

Line 448: "Significant at $p<0.05$ " This is repeated a few times in the text. Add just a single sentence at M&M section stating "statistical significance was defined at $\alpha=0.05$ " to cover all analyses.

Line 450: "Sub-sampling the OTUs counts for the lowest sample size". So the data were rarified? Was this done in R or in Qiime? What was the lowest sample size? This is important information.

Line 453: "Modified Gower distance" Which kind was used? Base 2 or base 10? I believe a phylogenetic beta-diversity measure would be most appropriate in this case (e.g. UniFrac).

Line 453-459: It is unclear why to me why statistical testing was executed this way. The effect of treatments should be tested by comparing their means, and this is done through ANOVA or PERMANOVA. Why was PERMANOVA selected here rather than ANOVA as was the case for alpha-diversity? As mentioned at line 148, PERMDISP will identify which groups have unequal variances, but this does not imply that the means of these groups differ. What should be included here are the pairwise comparisons of the means, i.e. output of pairwise.perm.manova or the pairwise.adonis2 function in R.

Line 492: Correct "polychaetes".

Line 493: "Trophic guilds". Explain in the same way as for nematodes in previous paragraph.

We would like to thank the three reviewers for their positive comments, stimulating feedback and the opportunity to address their constructive criticisms of our manuscript. Below, we have attended to all of the comments in a stepwise fashion and provided, what we hope to be comprehensive dialogue regarding their comments and our interventions in the revised manuscript.

Reviewers' comments:

Reviewer #1 (Remarks to the Author):

General

-Studies on meiobenthos are traditionally time-consuming and training requirements. Although the high-throughput sequencing (HTS) had been widely used in various fauna and microorganisms, it was rarely used in meiobenthos. The field experiment in Chwaka Bay had already been studied and published on the other themes (e.g. Dahl et al. 2016, Deyanova et al. 2017), and this study is focused on meiobenthos and the changes of their community structure after the seagrass treatments.

-Personally, I am not an expert of molecular biology so I am not familiar with the high-throughput sequencing (HTS) or related analyses. I would focus on the contents of meiobenthos and seagrass and their relationships.

Reply: We thank the reviewer for the insightful comments that helped us to improve the manuscript.

-The authors have good taxonomic data of nematodes from HTS, whereas there is no any detailed discussion about the alteration of nematode assemblages. For example, regarding Desmodorida, does Catanema in unvegetated site replace by Molgolaimus in other seagrass treatments? It seems like that it is not appropriate to overextend the meaning of OTUs, but the relative abundance of OTUs might still reflect some interesting variances in communities.

Reply: As pointed by Reviewer 3 (see below), differences between unvegetated and control areas were not the main focus of our paper. However, we agree with Reviewer 1 that a limited discussion about differences in nematode taxa would enrich the manuscript. As such, we have expanded the Discussion section to include this information in line 365-375.

-It does need some minor alterations. Please find my suggestion as the following.

Introduction

60 In De Troch et al. (2001), the “benthic” and “epiphytic” meiofauna actually indicate the meiobenthos associated with unvegetated and seagrass sediments, respectively. And meiofauna are usually considered identical to meiobenthos. Therefore, it better uses “epiphytic invertebrate biodiversity” here and need to cite the other references.

Reply: Changed as suggested by the reviewer in line 60-63. We have also changed the references to support this statement

78 *It might be better to add the family names of the fishes, e.g. emperors (Lethrinidae), for potential misunderstanding. This sentence may need to rewrite for easy reading.*
We added the family taxonomic name to the common name of the fish species
103 *...a decrease of seagrass above-ground biomass.*

Reply: We have rewritten the sentence and included the family names of the fish species mentioned in the text in line 77-80. The sentence now reads: “An additional important source of disturbance in seagrass beds comes from increased fishing pressure. The removal of predatory fishes such as wrasses (*Labridae*), snappers (*Lutjanidae*) and emperors (*Lethrinidae*)²⁹ can disturb the balance between herbivory and seagrass production and potentially induce cascading effects³⁰ in these ecosystems.”

Results

128 *If all the Mollusca in Supplementary Fig. 3-A belong to gastropods, it should denote Gastropoda.*

Reply :We changed Fig S3 (now Fig. 1 following Reviewer 2 comment) as suggested by the reviewer

Discussion

265 *Most nematodes might exhibit a certain level of generalist in feeding. However, their body sizes and sizes of buccal cavities restrict their food resources. The classification of Wieser (1953) still provide valuable information about the feeding guilds of nematode community.*

Reply : We have modified the sentence above to reflect the reviewers comment. It now reads in line 281-285: “Changes in nematode trophic structure should be interpreted cautiously as recent work suggest that most nematodes in their natural environment might exhibit a certain level of generalist and opportunistic feeding behavior⁴⁸. Nevertheless, the classification of Wieser (1953) still provides valuable information about the feeding guilds of nematode community.”

280 *It seems like that the feeding guilds of nematode and polychaete have slightly different responses to seagrass treatments. Of course, the feeding groupings may not truly represent their dietary habits, but some comparison between nematode and polychaete might be interested.*

Reply: The original submission included a paragraph where the response of polychaete feeding guilds were discussed. We have modified it to highlight the differences between nematode feeding groups and polychaete feeding guilds. It now reads in line 287-292: “Unlike what was seen with nematode feeding types, the abundance of predator polychaete was reduced in one our shading treatments. This is in accordance with previous studies that

have found an increase in dominance of polychaete deposit feeders and a decrease proportion of carnivores as an observed response to anthropogenic disturbance in benthic ecosystems^{49,50}.

311 *It is also possible that your clipping treatment not affect the below-ground community as severe as above-ground one. Cutting leaves and shoots would reduce the biomass of rhizomes, but the root and rhizome network still occurs.*

Reply: We added the additional explanation suggested by the reviewer regarding general effect of clipping. It now reads in line 332-335: “An additional explanation is that although simulated grazing treatments can reduce the biomass of rhizomes the root and rhizome network is still present and minimizes potential negative effects of above-ground disturbances on meiobenthic communities”

342 *About the effects of seagrass species' composition and density on meiobenthos. You may cite De Troch et al. (2001) and Liao et al. (2016).*

Liao, J.-X., Yeh, H.-M. & Mok, H.-K. Do the abundance, diversity, and community structure of sediment meiofauna differ among seagrass species? J. Mar. Biol. Assoc. U. K. 96, 725–735 (2016).

Reply: We have added the reference requested by the reviewer.

Methods

368 *Do the unvegetated replicate plots locate outside the experimental site? Since they are not indicated in Fig. 7, please declare in Methods.*

Reply : We have remade the figure (now Fig. 8) to include the unvegetated plots that were inside the experimental site adjacent to the remaining treatments. This information was also added 404-405.

388 *A surface area of 17 cm² is an appropriate size for sampling of meiofauna. Need references.*

Reply: We added two references to support this statement in line 428.

Fig. 5 and 6

The white asterisks showing significantly different are clear in the manuscript. However, it might be difficult to distinguish in a reduced-size copy. A traditional indication, e.g. asterisks beside the dots, may be appropriate here.

Reply: We have modified the figures (now Fig. 6 and 7) as suggested by the reviewer.

Reviewer #2 (Remarks to the Author):

Reviewer: Dr. Maickel Armenteros.

General comments

The manuscript addresses the ecological interactions between meiofauna and seagrass meadow features (perturbations) using an experimental approach. The research is pertinent and interesting to the generality of the marine ecology community working on meiofauna-diversity-ecosystem functioning. It covers a gap in the knowledge and provides novel and experimentally tested conclusions. Authors used a simple and well-designed experiment to test their hypotheses. They also applied modern molecular techniques to describe the meiofaunal communities. The manuscript is in general well written and focused on the goals. The statistical analysis is adequate for this sort of experimental work

I note that bioinformatics and HTS are far to be my expertise, so I did not focus on them. I present a set of criticisms and detailed comments below, but in general, positive features largely overcome the negative points, and I congratulate to the authors for so nice work.

Reply: We thank the reviewer for the positive words and the very useful comments during this round of revision that have improved the manuscript

Detailed comments

1. L1-3. *Trophic structure refers only to nematodes. I suggest "... meiobenthic diversity and nematode trophic structure"*.

Reply: We changed the title as suggested by the reviewer: "Above-below surface interactions in seagrass meadows: Indirect effects of disturbance on meiobenthic diversity and nematode trophic structure"

2. L19-20. *Abstract. There is some knowledge about impacts of seagrass degradation on E-F; I suggest modulating this statement since it is unfair.*

Reply: We have followed the reviewer's suggestion. This sentence in the abstract now reads: "Currently, we do not fully understand how seagrass habitat degradation impacts the biodiversity of belowground sediment communities".

3. L27. *Abstract. Change parameter x variable.*

Reply: Fixed

Introduction

4. L56. *Space between number and unit.*

Reply: Fixed

5. L77-83. *These two sentences are key, but as formulated hard to understand. I suggest rewriting, mentioning explicitly the grazers that increase activity when predator decrease. That is, more background information about the trophic links in the system.*

Reply: We have rewritten these two sentences and added the additional background information requested by the reviewer line 77-87. It now reads: "An additional important source of disturbance in seagrass beds comes from increased fishing pressure. The removal of predatory fishes such as wrasses (Labridae), snappers (Lutjanidae) and emperors (Lethrinidae)

²⁹ can disturb the balance between herbivory and seagrass production and potentially induce cascading effects ³⁰ in these ecosystems. Although, grazing is a vital process for controlling fast-growing epiphytic algae in eutrophic systems ³¹, release of grazers like sea urchins from predation can provoke intense grazing events that consume significant amounts of seagrass above-surface biomass ^{32,33}. High densities of sea urchins and consequent overgrazing of seagrasses have been more frequently reported in the last few decades ^{32,33} and can have enduring impacts on above-ground seagrass biomass ³², with potential important knock-on effects for sediment properties ³⁴ and the structure and function of benthic fauna communities.”

6. L95. *Change variable x treatment.*

Reply: We prefer the term variable in this context.

7. L99. *Please, rephrase “herbivore release from predation”.*

Reply: Changed as requested by the reviewer. In line 102-103 it now reads: “and simulated a high intensity grazing event due to herbivores being released from predation.”

8. L100-105. *It is supposed that all the background information was given previously. Is suggest remove references from hypotheses to easier reading.*

Reply: Changed as requested by the reviewer.

Results

9. L115-116. *I suggest rephrase as: Accumulation curves of number of OTUs versus number of samples for the six treatments.*

Reply: Changed as requested by the reviewer. In line 101-120 it now reads: “Accumulation plots of number of OTUs vs number of reads for each sample are presented in Supplementary Information (Supplementary Fig. 1).

10. *Supplementary fig. S1. Correct the caption. It is not a rarefaction curve. Rarefaction is a technique to compare the samples in reference to the smallest sample size. This is an accumulation curve of OTU vs. number of samples.*

Reply: Changed as requested by the reviewer.

11. L117. *I suggest that supplementary fig. S3 be included as a figure in the main manuscript and not as supplementary. It is relevant information about the results of the experiment and the ability of the HTS to describe the meiofauna community.*

Reply: We have moved Supplementary fig S3 to the main text and is now Fig. 1.

12. L122-123. *The word ‘remaining’ is repeated in same sentence, improve.*

Reply: We have rewritten this sentence line 127-128. It now reads: “The OTUs assigned to non-Metazoan Eukaryotes were excluded from the remaining analysis”.

13. *Supplementary fig. S3. Caption lacks important information. Indicate what are in the panels a, b, etc.*

Reply: This is now Fig.1 and we have included this information.

14. L125. Denote the table S1 as it. There are two excel files with cryptic names.

Reply: The excel files corresponding to Table S1 are labelled as Table S1. The file is supposed to be a read-only excel table, but it gets converted to a pdf file by the submission system. We have now submitted this as a data file which should solve the problem.

15. L128: 7 x seven.

Reply: Fixed.

16. L135. Alpha diversity has several definitions. I suggest to devote a couple of lines in the introduction to define what are the diversity metrics you are going to use, for inventory (~alpha), and variation (~beta) diversity components.

Reply: We have expanded the last paragraph of the Introduction section to introduce the components of diversity assessed in our study. Line 95-98.

17. L135. Repeated words, rephrase the sentence.

Reply: Fixed.

18. L137. I suggest using PERMANOVA instead of ANOVA for consistency. Former based on Euclidian distance is equivalent to ANOVA, it can be used in a univariate context as well, and is more robust to asymmetry and variance heterogeneity than ANOVA. Thus, you do not have to test for ANOVA assumptions (e.g. Bartlett test) and make transformations.

Reply: We have redone the statistics for alpha diversity measure using PERMANOVA as suggested by the reviewer. We updated the Results and Methods section with this information.

19. L146. Strictly speaking: ... the ordination of samples based on community structure...

Reply: We have modified the sentence as asked by the reviewer lines 157-158. It now reads: "Figure 3 shows an NMDS ordination of samples based on meiobenthic community structure across all treatments".

20. L147. PERMANOVA does not use F, instead pseudo-F.

Reply: Fixed.

21. L165. Ration x ratio.

Reply: Fixed.

Discussion

22. L345. I suggest adding the word summary or synthesis here. The relevance/transcendence of the study is here, and it is fine. But, I feel that some brief summary of the findings would improve this ending paragraph.

Reply: We have added a synthesis of our main results to open this paragraph in line 378-380. It now reads: "In summary, our results indicate that disturbance of seagrass meadows have

propagating effects on meiofauna community composition and nematode trophic structure, that are mediated by above-below ground interactions.”

M & M

23. L357. Year?

Reply: We have added this information in line 392.

24. L369. *The phrase four replicate plots should be in a separate sentence.*

Reply: Fixed.

25. L370. *It means that each plot is 3 m x 3 m? Are the plots square or rectangular (as in fig. 7)? Please, clarify.*

Reply: The plots were square and we have updated the figure (now Fig. 8) to reflect this.

26. L371-383. *I suggest to add in the fig. 7 some photos of the application of the treatment. E.g. How the deployed shading nets look? How look the seagrass when removed the blades? Maybe fig. 7 can be passed to supplementary.*

Reply: We have modified this figure (now Fig. 8) with pictures of how the seagrass meadows looked after the experimental treatments were applied. We think Fig.8 is useful for the reader to understand the design of the experiment, so we prefer to keep it in the main body of the text.

27. L444-446. *If you chose to use PERMANOVA, you do not have to use Bartlett test.*

Reply: We redone these tests with PERMANOVA as suggested by the reviewer and updated this section of the methods in line 486-487.

28. L447-448. *Mention Tukey post hoc test in other parts of the text instead of ANOVA (see comment above).*

Reply: See reply above.

29. L456-459. *Consider invert the order of these two sentences foe easier reading.*

Reply: We have inverted the order of these two sentences as suggested in line 493-498.

30. L487. *I suggest to change mouth x buccal cavity.*

Reply: Fixed.

Table

31. L716-718. *Table 1. Names of variables (e.g. sed., core) should be clarified in the heading (foot?) of the table.*

Reply: We have added the meaning of the environmental variables in the caption of Table 1.

Captions

32. L721-723. Caption fig. 1. Use metrics instead of measurement for consistency. C capital. Remove codes. Post hoc test is not an ANOVA, should be Tukey.

Reply: We have made the changes suggested by the reviewer, the caption of the figure (now Fig 2) reflects that the tests were done with PERMANOVA as suggested above.

33. L725. Caption fig. 2. The term “beta-diversity matrix” is confusing. Ordination of the samples is based on OTU matrix using certain similarity index.

Reply: Fixed.

34. L729. Caption fig. 3. Should be PERMDISP instead PERMANOVA.

Reply: Fixed.

35. L733-737. Actually, you represented here a biplot (samples + environmental “external” variables). I would remove CCA1 and CCA2. The phrase “... co-variant relationship between significant non-correlated environmental factors” is confusing; I suggest rewrite. Arrows are vectors representing the correlation between environmental variables and the axes.

Reply: We have rewritten the caption for this figure (now Fig 5).

36. Supplementary Fig S4. Sorensen.

Reply: Changed to Sørensen.

Figures

37. L764. Fig. 1. I suggest adding a measure of central position (e.g. mean or median) as a line (or bar), since statistical tests say differences in relation to mean (or centroid). It is matter of opinion, but I see too much blank space in the graphs (i.e. low information density). Larger size in letter and symbols would improve the plot.

Reply: We have remade this figure (now Fig 2) and have added a central bar representing the mean of each treatment. We have also increased the size of the symbols and letters in the plot.

Reviewer #3 (Remarks to the Author):

This manuscript (MS) provides experimental evidence to support the proposition that disturbance above the sediment surface in sea grass meadows generates indirect effects on meiobenthic diversity and trophic structure below/in the sediment. The authors focused on two types of disturbance, (i) reduced light availability (high/low) and (ii) grazing (high/low) with a 5-month (or week?) experiment, assessing the effect of the aforementioned treatments on alpha and beta diversity measures as well as community trophic structure of nematodes and polychaetes.

The main conclusions were that both treatments had an effect on meiobenthic communities, primarily at the beta-diversity level, nematode trophic structure (high shading/low clipping treatments) and polychaete community structure (low shade).

This MS represents a well-executed, balanced experiment with an original (to my, albeit, limited knowledge on the subject) perspective on linking disturbance above the sediment

surface to effects on infaunal taxa. The MS is generally well written, but will benefit from general grammatical improvements and increased clarity and conciseness.

Reply: We thank the reviewer for the positive words and the constructive comments and suggestions have contributed to improve the manuscript.

The application of high-throughput sequencing (HTS) adds to the value of the MS, however the adoption of OTUs rather than Amplicon Sequence Variants (DADA2) strikes me as disadvantageous given how the former are known to overestimate richness and are characterized by higher error rates in taxonomic assignments.

Reply: The use of ASVs has certainly advantageous aspects, some of them mentioned by the reviewer. However, it also comes with some important disadvantages, some of them relevant in the context of our study. Eukaryotes are known to be highly variable in the traditional target loci for molecular taxonomy with multiple divergent rRNA operons¹. As eukaryotic ASVs can differ by as little as 1 base pair², such intra-genomic heterogeneity can result in an overestimation of diversity. This because a population or species will have multiple ASVs attributed to itself. This problem is significantly reduced when using an OTU approach that clusters sequences up to a cutoff point. In addition, metabarcoding data often includes a number of PCR amplification and sequencing errors. The quality of the data becomes a bigger factor when working with ASVs, since often a large number of reads is lost in the denoising steps, as pipelines try to discriminate between such errors and real biological variation. An increase of single base pair errors can result in the removal of a larger number reads before downstream analyses when an ASV-based approach is used instead of OTUs. We argue that, while presenting an important set of advantages, an ASV approach is not necessarily always better than the more traditional OTU-based approach. The latter has been widely used in metabarcoding studies producing important advances in meiofauna diversity studies³⁻⁶.

The Materials & Methods section for Bioinformatics is lacking important parameter details. In addition, I would think that a beta-diversity measure that includes phylogenetic information (e.g. UniFrac) would be more appropriate given that sequence data are being analysed, compared to the chosen Gower distance.

Reply: We have included these details on the bioinformatics asked by the reviewer (see reply to the comments below) and analysed the sequence data with Unifrac distances (see a more detailed answer to a similar comment also below in the Line-by-line comments)

Many environmental variables were measured during these experiments (21 total), however we have no visual representation of their patterns in the different treatments. A figure of the ones that were most highly correlated with meiobenthos community composition at minimum (BIOENV analysis) would be appropriate as currently the reader has no way of knowing the general trends.

Reply: As requested by the reviewer, we have included an additional figure in supplementary information (Supplementary Fig.5) showing the variation among treatments of the seven variables found by the BIOENV analysis to be best correlated with meiobenthic community composition.

Line-by-line comments:

Line 24: “assessed” should be “assess”

Reply: Fixed.

Line 56: Meiobenthos definition by Giere actually extends to 1000µm. These days meiofauna is usually considered to lie in the 32-1000µm range.

Reply: Fixed.

Line 96: “We used shading to simulate the effects of reduced light availability to seagrasses at levels compatible with ecologically realistic effects of eutrophication and/or sedimentation [...]”. How was this ecological compatibility and relevance determined? Include reference if available with relevant irradiance levels, if not I would delete this sentence.

Reply: We have modified this sentence. It now reads line 100-103:” We used shading to simulate the effects of reduced light availability to seagrasses due to eutrophication and/or sedimentation, and simulated a high intensity grazing event due to herbivores being released from predation.”

Line 129: “Zooming in” change to “Within”

Reply: Fixed.

Line 130: “There were differences”. Were these statistically tested?

Reply: Differences in relative abundances of meiofauna taxa were tested with PERMANOVA. This is now mentioned in the Results (line 138-143), and in the Methods section of the revised manuscript.

Line 148: “A pairwise comparison performed with PERMDISP”. I am unsure if this is the most appropriate way to test the effect of treatments. This would be achieved through PERMANOVA, which requires a non-significant PERMDISP as an assumption (non-parametric equivalent of a homoscedasticity test). As far as I understand, PERMDISP will indicate which groups have unequal variance, but not which groups means differ, which is what you are trying to test. Non-parametric pairwise comparisons in R can be completed with pairwise.perm.manova or the pairwise.adonis2 function. As this is used throughout the text, its crucial to clarify what exactly is being tested.

Reply: We have clarified this section in the Results and Methods section. We tested if there was an effect of treatment on the average distance to group centroid with the *betadisp* function of the vegan package using a permutation test. This function implements Anderson et al.⁷ PERMDISP2 procedure. Pairwise differences between treatments were checked with the *permutest.betadisper* function using the *betadisp* object. This function permutes model residuals and generates a permutation distribution of F with the null hypothesis that there is no difference in dispersion between groups.

We tested if there an overall effect of treatment in community composition with PERMANOVA using the *adonis* function of the vegan package. The reviewer is correct that interpretation of PERMANOVA output with non-homogeneous dispersions is problematic, but for datasets with unbalanced sample sized among groups⁸. We had a balanced sample

size among groups (n=4 for all groups) so we can be confident in our *adonis* PERMANOVA results. PERMANOVA using *adonis* have been found to be largely unaffected by heterogeneity of dispersions according to Anderson & Walsh⁸. We followed the reviewers advice and used the *pairwise.perm.manova* function of the RVAideMemoire package to perform pairwise comparison between CTRL and the remaining treatments. We have modified the section in the Methods (line 493-504) and in the Results sections (line 158-175) clarifying what was done.

Line 150: "Average distance to centroid". Is this the Gower distance? If so, mention it by name.

Reply: Fixed.

Line 165: Correct "ration" to "ratio".

Reply: Fixed.

Line 172: "In regards to the nematodes" change to "With regards to the nematodes"

Reply: Fixed.

Line 224: "Both lower oxygen conditions ..." Include reference for this statement

We added two references to support this statement in line 242.

Line 226: "DOC". First time this acronym is encountered so should be written in full

Reply: Fixed.

Line 232: "In accordance, shading resulted ...". The way this sentence is phrased gives the impression to the reader that is referring to the current study, yet a reference is included, rephrase in a manner that makes it obvious you are referring to a different study, e.g. similar shading experiments have shown ...

Reply: We have rewritten this sentence. It now reads in line 250-253: "Similar in situ studies have shown that shading resulted in a significant decrease in root biomass and photosynthetic activity in the HS treatments 25 and the BIOENV analysis in our study identified rhizome biomass as one of the variables that correlated with meiobenthic beta-diversity".

Line 243: "Foods sources available to the nematodes" change to "food availability" to avoid repeating "nematodes" twice in one sentence.

Reply: Fixed.

Line 248: "Functional classes" or feeding types? Both have been used in the MS, chose one for consistency.

Reply: We changed functional classes to feeding types for consistency.

Line 250: "We propose that..." This sentence needs restructuring for clarity.

Microphytobenthos and diatoms are both phytoplankton so, if I understand correctly, essentially the message is that shading reduced phytoplankton production and therefore epistrate-feeding nematodes. E.g.: We propose that shading reduced important phytoplankton food sources to epistrate feeder nematodes as well as sedimentation in these plots, thereby decreasing the relative abundance of epistrate feeders.

Reply: We clarified this sentence as suggested by the reviewer in 269-271.

Line 255: "NCP was in fact significantly ..." Here again the phrasing gives the impression that the conclusion is made from the current study data, but a reference is provided. Rephrase to clarify which is true.

Reply: We have rewritten this sentence. It now reads in line 272-274: "A study that used the same experimental system as ours found community metabolism to be significantly lower in the HS treatment than in the CTRL plots³¹."

Line 258: "The different food sources" Which ones? Or just different? Rephrase sentence for clarity

Reply: It is mentioned in that sentence which food sources: "nematodes classified as selective deposit feeders are generally considered to depend on different food sources than epistrate feeders, as they preferentially feed on bacteria, small particulate food or dissolved organic matter".

Line 266-268: "Nevertheless, because of its easy application" You just mentioned that the feeding types should be interpreted with caution and that's sufficient in my view, I would delete this entire sentence justifying using Wiesers scheme.

Reply: We have modified this sentence as also suggested by reviewer 1 in line 284-285: It now reads: "Nevertheless, the classification of Wieser (1953) still provide valuable information about the feeding guilds of nematode community."

Line 276: "Nematode community" But also polychaetes right?

Reply: Correct, we have added polychaetes 1 in line 295.

Line 289: "Indeed, this was reflected ..." Here again the phrasing gives the impression that the conclusion is made from the current study data, but a reference is provided. Rephrase to clarify which is true.

We have rewritten this sentence. It now reads in line 308-310: "Indeed, Dahl et. al³¹ found a lower organic carbon content in the first 2.5 cm layer of sediment of the clipping treatments in the same experimental system here reported."

Line 301: "As a result from seagrass biomass" change to "as a result of seagrass biomass"

Reply: Fixed.

Line 320: “HC treatment compared to the CTRL”. Here again the phrasing gives the impression that the conclusion is made from the current study data, but a reference is provided. Rephrase to clarify which is true.

We have rewritten this sentence. It now reads in line 337-339: “We also anticipated changes in sediment condition in the HC to affect the trophic structure of nematodes; in particular the abundance of OTUs of epistrate feeders as Dahl et al.³¹ found significantly higher Chla content in HC sediments.”

Line 328-344: I think it is worth mentioning that sea grass beds can have a negative effect on meiofauna abundance, and that these effects may be species-specific. However, the MS is aimed at demonstrating the effects of the experimental treatments, for which the CTRL site is relevant. Highlighting differences between CTRL and unvegetated seems to me irrelevant here, and also rather obvious that these would differ.

Reply: As stated both in the Introduction and Discussion sections, studies investigating effects of seagrasses on meiobenthic communities have shown contrasting results. We feel that, although this was not the principal aim of our study, our results contribute with relevant information to this debate as we found clear negative quantitative effects on meiobenthic diversity. In addition, Reviewer 1 asked to expand this section to include some discussion on differences between nematode taxa. We have therefore kept this part of the discussion in the revised manuscript. We have, nevertheless, tried to keep this section short to accommodate Reviewer 3 comment.

We mentioned that seagrasses can have negative effects on meiofauna abundances in the previous submission: “Furthermore, a number of studies have shown meiobenthos abundance to be negatively correlated to seagrass cover as a result of increased predation pressure by macrofauna on vegetated sediments^{56,57}”.

Line 370: Fig.7 illustrates 5 of the 6 treatments, where are the unvegetated blocks located?

Reply : We have remade the figure (now Fig. 8) to include the unvegetated plots.

Line 373: “Mean light reduction over day of 64% or 307 $\mu\text{mol quanta m}^{-2} \text{s}^{-1}$ ” Would be more comprehensible for the reader to state what the CTRL irradiance is first (470), and then to what irradiance the LS and HS will be. LS: $470-307=163$ final irradiance or reduction of 65% ($307/470=65.32\%$) | HS: $470-356=114$ final irradiance or reduction of 75%

Reply : We have rephrased this sentence. It now reads in line 410-413: “This procedure reduced the light irradiance from $470 \mu\text{mol quanta m}^{-2} \text{s}^{-1}$ in the seagrass control plots, to $356 \mu\text{mol quanta m}^{-2} \text{s}^{-1}$ in the LS treatment (a mean light reduction over day of 64% in relation to CTRL) and $307 \mu\text{mol quanta m}^{-2} \text{s}^{-1}$ in the HS treatment (a mean light reduction over day of 75% in relation to CTRL).”

Line 386: “After 5 weeks” Should be 5 months correct?

Reply: Yes, the reviewer is right. This was corrected.

Line 387: “Sediment cores taken from the same points inside the plots” What does this mean?

Reply: This sentence was clarified. It now reads in line 424-426: “After 5 months at the termination of the experiment the sediment of each of the 24 replicate plots was sampled with six handheld Perspex sediment cores taken from the exact same location within each of the plots”

Line 393: Delete 40µm, mentioned in previous sentence. “In situ salinity” Which was what? Measured how?

Reply: Salinity at the site varied between 33-34 throughout the experiment and it was measured with a multimeter Multi 340i, CelloX 325 (WTW). We added this information in line 401.

The water used to wash the DESS was prepared in the lab with artificial water at salinity 34. We clarified this in the text in line 430-432. It now reads: “After two weeks the sediment and animals were again placed in a 40 µm sieve and rinsed thoroughly in filtered artificial saltwater (salinity 34) close to in situ salinity to remove the DESS”.

Line 403: “Volume of water” What kind of water? Care was taken to keep the sieves sterile, but frequently water can be the source of contamination in samples. Did you use Milli-Q water as the final solution for the extracted animals?

Reply: We did indeed use Milli-Q water for this part. We added this in the text in line 441.

Line 415: “Each sample from the 24 replicate plots” If I understand correctly, each of the 24 pools includes the triplicates of each of the 6 replicate cores per plot? Difficult for the reader to understand what happened to those 6 replicate cores, include this info for clarity.

Reply: The 6 replicate cores of each plot we pooled before sieving as stated in line 390-391 (now line 429). 24 DNA extractions were performed. Each PCR reaction for library preparation was done in triplicates, then pooled after cleaning with Agencourt AMPure XP PCR Purification (line 455). The sentence in line 420-424 (now line 459-463) further clarifies how many samples were sequenced.

Line 429: “Quality filtering”. What was the chosen phred-score threshold? This is important information.

Reply: A minimum Phred quality score of 19 was used. We added this information in line 469.

Line 432: “Trimmomatic” Again, what were the chosen parameters? This is important information.

Reply: Parameters used for Trimmomatic were ILLUMINACLIP:2:30:10, with all other parameters as default. We added this information in line 472.

Line 437: “RDP Classifier” What was the chosen confidence threshold? This is important information.

Reply: We used a with a confidence threshold of 0.7. We added this information in line 478.

Line 445: Was data normality tested? This should also be checked.

Reply: This sentence was removed after following the suggestion of reviewer 2 to redo these analyses with PERMANOVA.

Line 446: Delete “to examine for all ANOVA analysis”, mentioned at beginning of sentence

Reply: This sentence was removed after following the suggestion of reviewer 2 to redo these analyses with PERMANOVA.

Line 448: “Significant at $p < 0.05$ ” This is repeated a few times in the text. Add just a single sentence at M&M section stating “statistical significance was defined at $\alpha = 0.05$ ” to cover all analyses.

Reply: We have modified this sentence as suggested by the reviewer (line 487).

Line 450: “Sub-sampling the OTUs counts for the lowest sample size”. So the data were rarefied? Was this done in R or in Qiime? What was the lowest sample size? This is important information.

Reply: We have included this information in line 488-490. The sentence now reads: “Community composition was examined by first selecting and filtering metazoan OTUs and sub-sampling the OTUs counts to the lowest sample size (66 754 counts) with the `rarefy_even_depth` function in `pyloseq`”

Line 453: “Modified Gower distance” Which kind was used? Base 2 or base 10? I believe a phylogenetic beta-diversity measure would be most appropriate in this case (e.g. UniFrac).

Reply: We used the "altGower" which divides all distances by the number of pairs with at least one above-zero value, without scaling columns to its range or range standardization⁷. We have clarified this in the text in line 492.

As suggested by the reviewer we have performed a new ordination with a Principal Coordinates Analysis (PCoA) with UniFrac distance. The results were very similar both visually and for the statistics (*adonis*, Pseudo- $F_{5,18} = 4.6$, $p = 0.001$). Since the results were similar and NMDS ordinations is a common ordination for these types of studies, we decided to keep the NMDS in the main file and include the PCoA with Unifrac distances as Supplementary Information

Line 453-459: It is unclear why to me why statistical testing was executed this way. The effect of treatments should be tested by comparing their means, and this is done through ANOVA or PERMANOVA. Why was PERMANOVA selected here rather than ANOVA as was the case for alpha-diversity? As mentioned at line 148, PERMDISP will identify which groups have unequal variances, but this does not imply that the means of these groups differ. What should be included here are the pairwise comparisons of the means, i.e. output of `pairwise.perm.manova` or the `pairwise.adonis2` function in R.

Reply: We have now standardized the tests, PERMANOVA was also used for alpha diversity (See reply to Reviewer 2). We have now clarified the statistics in the methods section (see reply above to Reviewer 3's that starts with line 148).

Line 492: Correct “polychaetes”.

Reply: Fixed.

Line 493: “Trophic guilds”. Explain in the same way as for nematodes in previous paragraph.

Reply: We added this information in line 538-539.

References

1. Bik, H. M. *et al.* Sequencing our way towards understanding global eukaryotic biodiversity. *Trends Ecol. Evol.* **27**, 233–43 (2012).
2. Callahan, B. J. *et al.* DADA2: High-resolution sample inference from Illumina amplicon data. *Nat. Methods* **13**, 581–583 (2016).
3. Fonseca, V. G. *et al.* Second-generation environmental sequencing unmasks marine metazoan biodiversity. *Nat. Commun.* **1**, 98 (2010).
4. Lallias, D. *et al.* Environmental metabarcoding reveals heterogeneous drivers of microbial eukaryote diversity in contrasting estuarine ecosystems. *ISME J.* (2014). doi:10.1038/ismej.2014.213
5. Schuelke, T., Pereira, T. J., Hardy, S. M. & Bik, H. M. Nematode-associated microbial taxa do not correlate with host phylogeny, geographic region or feeding morphology in marine sediment habitats. *Mol. Ecol.* **27**, 1930–1951 (2018).
6. Leray, M. & Knowlton, N. DNA barcoding and metabarcoding of standardized samples reveal patterns of marine benthic diversity. *Proc. Natl. Acad. Sci. U. S. A.* **112**, 2076–81 (2015).
7. Anderson, M. J., Ellingsen, K. E. & McArdle, B. H. Multivariate dispersion as a measure of beta diversity. *Ecol. Lett.* **9**, 683–693 (2006).
8. Anderson, M. J. & Walsh, D. C. I. PERMANOVA, ANOSIM, and the Mantel test in the face of heterogeneous dispersions: What null hypothesis are you testing? *Ecol. Monogr.* **83**, 557–574 (2013).

REVIEWERS' COMMENTS:

Reviewer #1 (Remarks to the Author):

The revised MS is well written and can be accepted.

With some minor corrections:

The family or order names should not be italicized. Please check Line 78, 79, 141–143, 371 and 377.

Add labelling in Fig. 8 (B) and (C).

Reviewer #2 (Remarks to the Author):

General comment

Authors have made a great work that have improved the previous version of the manuscript.

Therefore, the novelty, quality and relevance of the manuscript are now better addressed. Here I have a group of additional remarks. I recommend the manuscript be published after the correction of these new remarks.

Abstract. OK.

Introduction

- L57. Meiobenthos not only live in sediments, also phytal and hard-bottom (consolidate) substrates. Please, modify.
- L98. Objective iii. The trophic structure of polychaete assemblage s is also addressed, maybe it should be reflected in the title, besides of nematodes.
- Please, check for consistency that the two treatments (shading and clipping) be mentioned in the same order.

Results

- L124. As M&M section is placed at the end, you should spell out here the meaning of the treatment abbreviations (e.g. HC, HS, etc.).
- L139. Nematode twice, remove one.
- L145. No description of the pattern to genus level (Fig. 1C). You should say something here, or remove the panel 1C.
- L148. As usually, Shannon index does not says nothing more than the other used metrics (Unique OTUs and ACE). Consider to remove for simplicity and since you do not mention it again.
- L172. One parenthesis missing.
- L182. Parameter refer to the modelling jargon (e.g. intersect and slope are parameters in a model), I recommend to use variable that refer to quantities you have measured. I do not see the needed to use also factor ... Use only variable for simplicity. Also check in M&M (e.g. L525).
- L192. Consider include polychaetes in the title.

Discussion

- I think that hypotheses should conduct the discussion. The two first subsections (effects of shading ... and grazing ...) should be linked to the proposed hypotheses and explicitly says if they were rejected or not.
- L363. Meadow.
- L371-381. I recommend to revise this analysis at order level. Usually the signals at order-level are elusive or even absent. For instance, desmodorids bear a wide spectrum of cuticle type, from very finely annulated (e.g. Catanema) to coarse annulation (e.g. Croconema or Zalonema). In particular, I doubt that the fine cuticle of Catanema plus its ectosymbiotic bacterial layer enhances locomotion.

- L383. No effects on polychaete trophic structure?
- L385. You have used seagrass alone, bed, and meadow. I recommend to use only one term.
- L382-393. Summary paragraph would benefit of a better organization. The scheme in the abstract, although short, looks fine: effects on meiobenthos along with drivers, effects on trophic groups, and expanding the finding to wider scope. You may follow in the summary the same scheme with more details. Also, refer to the working hypotheses, they gave the appropriate framework for the research.

Materials and methods

- L511. Sample volume. Is this correct? At least, unclear for me.
- L516. Remove "method".
- L517. BIOENV indicates the subset that best "correlate" to the similarity biotic pattern; and after, we assume that this subset best "explain" the pattern. It is not the same.

Figure

- L815. Consider change "sampling point" x data point or observation or sample; also in L505.

We would like to thank again the editor and the three reviewers for their positive feedback that helped to greatly improve of our manuscript. Below, we have attended to all of the reviewers comments and delivered an enhanced revised version of our manuscript.

REVIEWERS' COMMENTS:

Reviewer #1 (Remarks to the Author):

The revised MS is well written and can be accepted.

With some minor corrections:

The family or order names should not be italicized. Please check Line 78, 79, 141–143, 371 and 377.

Reply: Corrected as suggested by the reviewer

Add labelling in Fig. 8 (B) and (C).

Reply: We added the panel labels on Fig.8

Reviewer #2 (Remarks to the Author):

General comment

Authors have made a great work that have improved the previous version of the manuscript. Therefore, the novelty, quality and relevance of the manuscript are now better addressed. Here I have a group of additional remarks. I recommend the manuscript be published after the correction of these new remarks.

Abstract. OK.

Introduction

- L57. Meiobenthos not only live in sediments, also phytal and hard-bottom (consolidate) substrates. Please, modify.

Reply: We have rewritten this sentence: It now reads: "The abundance and diversity of below-surface metazoan consumers in marine sediments is dominated by meiobenthos (microscopic benthic invertebrates between 0.04 and 1 mm in size)"

- L98. Objective iii. The trophic structure of polychaete assemblage s is also addressed, maybe it should be reflected in the title, besides of nematodes.

Reply: We have included polychaetes in the title

- Please, check for consistency that the two treatments (shading and clipping) be mentioned in the same order.

Reply: We doubled in line checked that these treatments are mentioned in consistent order throughout the Introduction.

Results

- L124. As M&M section is placed at the end, you should spell out here the meaning of the treatment abbreviations (e.g. HC, HS, etc.).

Reply: We added the meaning of the treatment abbreviations as suggested by the reviewer in line 125-126.

- L139. Nematode twice, remove one.'

*Reply: Corrected as suggested by the reviewer, it now reads:
"Within the nematodes, there were differences among treatments in relative abundances of its taxa".*

- L145. No description of the pattern to genus level (Fig. 1C). You should say something here, or remove the panel 1C.

Reply: We have added to the Results section a description of the most important differences in relative abundances of nematode genus that connect to the Discussion. It now reads in lines 145-150: "At the nematode genus level there was a conspicuous difference in dominance between the seagrass plots (CTRL, HS, LS, HC and LC) and the Unvegetated treatments. While the former were dominated by Molgolaimus and Monhysterids nematodes (PERMANOVA, pseudo- $F_{5,18} = 6.1$, $p = 0.002$, pseudo- $F_{5,18} = 29$, $p = 0.001$, respectively) the latter treatment was dominated by nematodes of the genus Catanema (PERMANOVA, pseudo- $F_{5,18} = 64.3$, $p < 0.001$ - Fig. 1-C)"

- L148. As usually, Shannon index does not says nothing more than the other used metrics (Unique OTUs and ACE). Consider to remove for simplicity and since you do not mention it again.

Reply: Shannon index in a common alpha diversity metric in eukaryotic metabarcoding studies. We think that different metrics that represent different aspects of alpha diversity show the same pattern in our study is a strength that is informative and useful for the reader. As such we prefer to retain the panel of the figure that shows Shannon diversity.

- L172. One parenthesis missing.

Reply: Corrected

- L182. Parameter refer to the modelling jargon (e.g. intersect and slope are parameters in a model), I recommend to use variable that refer to quantities you have measured. I do not see the needed to use also factor ... Use only variable for simplicity. Also check in M&M (e.g. L525).

Reply: As suggested by the reviewer, we replaced the words "parameters" and "factor" for "variables" both in the Results and Material and Methods section

- L192. Consider include polychaetes in the title.

Reply: The title of this section now reads: "Differences among treatments in trophic composition of nematodes and polychaetes"

Discussion

- I think that hypotheses should conduct the discussion. The two first subsections (effects of shading ... and grazing ...) should be linked to the proposed hypotheses and explicitly says if they were rejected or not.

Reply: We think that the two subsections of the Discussion were already linked to the Introduction. Each of the two hypothesis outlined in the Introduction were addressed in the correct order in each of the two sub-sections of the Discussion. We specifically address the hypothesis in the text of the Discussion. Hypothesis one (outline in line 102-104) is explicitly addressed in lines 298-300:

"Taken together our results clearly show an indirect effect of shading on meiobenthic community composition and trophic structure that is mediated by seagrass response to eutrophication/and or increased sedimentation".

Hypothesis two (line 104-107) was less explicitly addressed in the Discussion. We have improved this in lines 349-352. It now reads: "Although an effect of clipping was detected on meiobenthic beta-diversity, community composition and nematode trophic structure, our results indicate that disturbance related to clipping has less pronounced effects when compared to shading"

- L363. Meadow.

Reply: Typo corrected

- L371-381. I recommend to revise this analysis at order level. Usually the signals at order-level are elusive or even absent. For instance, desmodorids bear a wide spectrum of cuticle type, from very finely annulated (e.g. *Catanema*) to coarse annulation (e.g. *Croconema* or *Zalonema*). In particular, I doubt that the fine cuticle of *Catanema* plus its ectosymbiotic bacterial layer enhances locomotion.

Reply: The analysis of differences in abundances at the order and genus level was requested by reviewer 1 in the previous revision. We do, nevertheless, acknowledge the reviewers concerns and have revised and toned down our discussion in this section of the text. It now reads:

*"Our results suggest that this habitat modulation by seagrasses influenced nematode community composition. Unvegetated sediments were dominated by Desmodorida, particularly of the genus *Catanema* that seem to find unstable fluid sediments in unvegetated areas advantageous^{14,18}. However, other studies have found *Catanema* to be common in seagrass areas at sediment depths deeper than the ones sampled in our experiment^{18,19}. *Catanema* was replaced by *Molgolaimus* in our seagrass plots, a common nematode genus in sediments of *T. hemprichii* meadows, particularly in its top layer 18. These seagrass plots were clearly dominated by Monhysterida, which are likely positively impacted by increased amounts of fine particles and detritus normally found in sediments in seagrass meadows⁶³. Effects of seagrass on nematodes and other meiobenthos may, nevertheless, be dependent on seagrass species' composition and density and on other abiotic factors not examined here.*

- L383. No effects on polychaete trophic structure?”

Reply: We have added polychaete to the sentence. We agree with the reviewer it reflects better our results

- L385. You have used seagrass alone, bed, and meadow. I recommend to use only one term.

Reply: We think alternating between these terms, that are essentially synonyms, prevent the text from becoming too repetitive and improves its flow for the reader.

- L382-393. Summary paragraph would benefit of a better organization. The scheme in the abstract, although short, looks fine: effects on meiobenthos along with drivers, effects on trophic groups, and expanding the finding to wider scope. You may follow in the summary the same scheme with more details. Also, refer to the working hypotheses, they gave the appropriate framework for the research.

Reply: We have rewritten this paragraph to accommodate the reviewers suggestions. We tried to follow the recommended structure and made a direct connection to the hypothesis stated in the Introduction section. This last paragraph now reads:

“In summary, our results indicate that disturbance of seagrass meadows have propagating effects on meiobenthic communities that are mediated by above-below ground interactions. Shading altered meiobenthic community composition and nematode and polychaete trophic structure to a larger dominance of deposit feeders. Such responses to shading by the meiobenthos seem to be related to reduced seagrass root and rhizome biomass reported in previous studies^{28,34}. The continued grazing in the clipping treatments also resulted in significant changes in meiobenthic community and similar changes in trophic structure, although these were not as clear as the shading treatments. Our study suggests that such changes are connected to a decrease in above-ground biomass and intensified erosion of the sediment surface reported in previous work³⁴. Since human-induced disturbances are increasing the rate of seagrass bed habitat degradation⁶³ it is crucial to improve our understanding of what such losses mean for the structure and functioning of benthic ecosystems. Our results highlight the complex role of above-below ground interactions in marine systems. Seagrasses function as ecosystem engineers for benthic faunal communities, and how they respond to disturbances can have significant indirect effects of meiobenthic community diversity and trophic structure. Considering that meiobenthos can have important roles in benthic foodwebs^{10,35} and mediate vital benthic ecosystem function^{11,13}, prolonged disturbances of seagrass habitats as presently seen in many coastal waters, are likely to have important cascading effects for benthic ecosystem structure and function.”

Materials and methods

- L511. Sample volume. Is this correct? At least, unclear for me.

Reply: It should read treatments. We have corrected this and thank the reviewer for spotting this mistake

- L516. Remove “method”.

Reply: Removed as suggested by the reviewer

- L517. BIOENV indicates the subset that best “correlate” to the similarity biotic pattern; and

after, we assume that this subset best “explain” the pattern. It is not the same.

Reply: We have replaced the word “explain” with “correlated with” in this sentence.

Figure

- L815. Consider change “sampling point” x data point or observation or sample; also in L505.

Reply: We have replaced “sampling point” with “observation as suggested by the reviewer both in line 505 and 815.

Reviewer #3 (Remarks to the Author):

Above-below surface interactions in seagrass meadows: Indirect effects of disturbance on meiobenthic

diversity and nematode trophic structure

Francisco J.A. Nascimento, Martin Dahl, Diana Deyanova, Liberatus D. Lyimo, Holly M. Bik, Taruna Schuelke³, Tiago José Pereira, Mats Björk, Simon Creer, Martin Gullström

I commend the authors on the adjustments and substantial improvements made to the MS, especially in such little time. Below you can find my suggestions.

A final spell check should be made to the MS as there are a few instances of small typos throughout the text (mentioned below).

Reply: We thank the reviewers for their contributions with this matter. We have made one final spell check on the manuscripts.

Lastly, I would advise the authors to include the statistical analyses outputs as Tables in the Supplementary Material.

Reply: We have now made a table with the statistical analysis outputs as suggested by the reviewer. It is included in the new submission as Supplementary Data 3.

Revised manuscript review

Line103: “Herbivory was simulated by clipping of shoots to simulate two different levels of grazing pressure”

Double use of “simulate” in a single sentence: Grazing by herbivores was simulated by clipping of shoots at two intensity levels” or similar.

Reply: We have rewritten this sentence. It now reads: “Herbivory was simulated by clipping of shoots to mimic two different levels of grazing pressure”

Line470: “...minimum Phred quality score of 19.”

I find this to be a rather lenient threshold, most studies I have read set this to 30 at least, or even 35. A Phred score of 19 represents an error rate of 1.259%, meaning that each 365-410 bp read will have at least 3-4 erroneous base calls, with quite a cumulative affect across the millions of reads generated. I would advise to increase this threshold to 30 at minimum. I understand this can be a major setback in the analysis (which I have experienced myself for

the same reason), nonetheless, it is best to be extra stringent in order to be able to be confident regarding your dataset. (https://wiki.bits.vib.be/index.php/File:Phred_to_probability.png)

Reply: The phred score used in our study followed the recommendations and guidelines of the Earth Microbiome Project ¹ (<http://www.earthmicrobiome.org/protocols-and-standards/initial-qiime-processing/>), set based on Bokulich et al ² work.

The 1% error rate that the phred score permits is four times smaller than the 96% clustering (i.e. 4% distance) we implement for OTU delimitation, meaning that reducing this further will have a negligible effect on the biodiversity metrics featured in the study. The errors in the biodiversity reads will only effect within and between closely related OTUs - not become additive throughout the whole study. In our combined experience, tweaking the dynamic interplay between quality filtering and OTU clustering only ever changes precise OTU richness (alpha diversity) values, whilst not altering emergent ecological responses or shifts in beta diversity which is the focus of our study. Nevertheless, to minimize such effects we used a OTU table that had a minimum cluster size of 2, meaning that we additionally filtered out all singleton OTUs which would have further filtered out low-quality Illumina sequences that remained in our dataset. This same approach with the same phred score and cluster size has been used in multiple studies that used the QIIME pipeline to investigate meiofauna diversity eg: Lallias et al ³, Taruna et al. ⁴, Nascimento et al. ⁵, Lobo et al. ⁶. Additionally, multiple other publications that studied the diversity and community structure of other taxa with Illumina amplicon sequencing have used the same parameter (see full publication list in: <http://www.earthmicrobiome.org/publications/>).

QA/QC thresholds in metabarcoding studies can vary quite substantially depending on lab and investigator preferences, and we inferred our quality threshold using best practices recommended in the peer-reviewed literature ². Since these practices are thoroughly tested and implemented with robust results in many other publications, and that an increase in phred score is unlikely to change our results, we argue that it would be unwarranted to redo the whole data analysis at this stage of the reviewing process.

Line545-547: “Differences in OTU abundance for nematode feeding groups and polychaete feeding guild between treatments were considered significant at a $p < 0.05$. Differences among treatments were tested with a Tukey HSD posthoc test for all variables and were considered significant at a $p < 0.05$.” You have mentioned that significance was set at $\alpha = 0.05$ at line 488, this does not have to be repeated twice here again.

Reply: We have revised this section of the text as suggested by the reviewer. We have removed the sentences on the α level at which differences were considered statistically significant, and mention only what tests were used and why they were used. It now reads:

“To assess differential OTU abundance between the CTRL and the other treatments in nematode and polychaete trophic structure, we used the DESeq2 statistical package ⁸⁶. DESeq2 accounts for the variance heterogeneity often observed in sequence data by using a negative binomial distribution as an error distribution to compare abundance of each OTU between groups of samples ⁸⁶.”

Line249: “... these direct and indirect change ...”
Correct to “changes”

Reply: Corrected as suggested by the reviewer

Line282: “... recent work suggest ...”

Correct to “suggests”

Reply: Corrected as suggested by the reviewer

Line288: “... the abundance of predator polychaeta ...”
Correct to “polychaetes”

Reply: Corrected as suggested by the reviewer

References

1. Thompson, L. R. *et al.* A communal catalogue reveals Earth’s multiscale microbial diversity. *Nature* **551**, 457–463 (2017).
2. Bokulich, N. A. *et al.* Quality-filtering vastly improves diversity estimates from Illumina amplicon sequencing. *Nat. Methods* **10**, 57–59 (2013).
3. Lallias, D. *et al.* Environmental metabarcoding reveals heterogeneous drivers of microbial eukaryote diversity in contrasting estuarine ecosystems. *ISME J.* (2014). doi:10.1038/ismej.2014.213
4. Schuelke, T., Pereira, T. J., Hardy, S. M. & Bik, H. M. Nematode-associated microbial taxa do not correlate with host phylogeny, geographic region or feeding morphology in marine sediment habitats. *Mol. Ecol.* **27**, 1930–1951 (2018).
5. Nascimento, F. J. A., Lallias, D., Bik, H. M. & Creer, S. Sample size effects on the assessment of eukaryotic diversity and community structure in aquatic sediments using high-throughput sequencing. *Sci. Rep.* **8**, 11737 (2018).
6. Lobo, J., Shokralla, S., Costa, M. H., Hajibabaei, M. & Costa, F. O. DNA metabarcoding for high-throughput monitoring of estuarine macrobenthic communities. *Sci. Rep.* **7**, 15618 (2017).